# Wildfire-Atmosphere Interaction Index for Extreme Fire behaviour

Tomàs Artés[1], Marc Castellnou[2], Tracy Houston Durrant[3], and Jesús San-Miguel[1]

[1]European Commission, Joint Research Centre (JRC), Ispra, Italy
[2]Grup de Recolzament a Actuacions Forestals (GRAF), Generalitat de Catalunya, Carretera de l'Autònoma, sn. 08290 Cerdanyola del Vallés, Spain
[3]External consultant for the European Commission (Engineering Ingegneria Informatica S.p.A.) Engineering Ingegneria Informatica S.p.A., Rome, Italy

**Correspondence:** Tomàs Artés (tomas.artes-vivancos@ec.europa.eu)

**Abstract.** During the last 20 years extreme wildfires have challenged firefighting capabilities. Often, the prediction of the extreme behaviour is essential for the safety of citizens and fire fighters. Currently, there are several fire danger indices routinely used by firefighting services, but they are not suited to forecast extreme wildfire behaviour at global scale. This article proposes a new fire danger index, the extreme fire behaviour index (EFBI), based on the analysis of the vertical profiles of the atmosphere above wildfires as an addition to the use of traditional fire danger indices. The EFBI evaluates the ease of interaction between wildfires and the atmosphere that could lead to deep moist convection, erratic and extreme wildfires. Results of this research through the analysis of some of the critical fires in the last years show that the EFBI can potentially be used to provide valuable information to identify convection-driven fires and to enhance fire danger rating schemes worldwide.

## 1 Introduction

Fires have naturally occurred in nearly all world biomes, shaping ecosystems and landscapes but are intrinsically linked to human activities. As humans spread to colonize most regions of the world, they brought fire with them and used it as a tool in agriculture and cattle raising activities. Currently, over 90% of the fires that occur in the world are caused by humans, deliberately or accidentally (Balch et al., 2017; Short, 2017; San-Miguel-Ayanz et al., 2012). In most regions of the world the most prominent fire management policy is that of fire exclusion. Fires that are purposely or accidentally started affect human assets and are thus controlled and extinguished as fast as possible. Only fires in very high latitudes, where human dwellings do not exist, are left to burn naturally. In the context of this paper we use the term wildfires, referring to those fires that escape beyond human control and cause damage to human lives and properties. Nowadays, it is estimated that about 400 million ha of natural and agricultural lands are burnt annually, although it is recognized that this figure is likely a gross underestimate of the total area burnt in reality (Boschetti et al., 2019). Wildfires are responsible for vast economic and environmental damage, the loss of human lives and about 17% of the global $CO_2$ (Shi et al., 2015; Friedlingstein et al., 2019) emissions. Most wildfires occur in the vicinity of humans, as they are caused by humans, and thus they affect people and human assets in the area where they occur. In Europe, this inter-mingle of human dwellings and natural areas is referred to as the wildland urban interface(WUI) and corresponds to the area where most fires and burnt areas occur.

The damage caused by fires in the WUI is thus much greater than that of fires occurring in remote areas due to the high adaptability of ecosystems in those areas(Pausas and Keeley, 2009, 2017). In recent years, the occurrence of extreme fire seasons has increased dramatically in many regions of the world, being associated in most cases with the effect that climate change is already posing on wildfire regimes and wildfire behaviour. Examples of these extreme wildfires were those occurring in Indonesia (2015), Chile (2016), California (USA, 2017, 2018,2020), Canada (2017), Portugal (2017), California (2018), Greece (2018), Australia (2019), Siberia (2019), Argentina (2021), Brazil, Bolivia and Paraguay (2019). Common to these events was the explosive behaviour of the single fires, which resulted in the loss of many lives and huge economic damage. Several entities reported and analysed these wildfires that interact with the atmosphere(Delicado and Gomes; Guerreiro, 2017) increasing wind speed, causing sudden wind direction changes, local fire tornadoes(Lareau et al., 2018b; Pirsko et al., 1965) and vortexes(Khaykin et al., 2020).

Often, the fire management agencies in the countries make use of past fire history to analyse the potential behaviour of future wildfires. Fires are characterized by their rate of spread and fire intensity as well as the weather and fuel conditions when they occurred. One of the most commonly used fire danger indices is the Canadian Fire Weather index (Van Wagner et al., 1974); this is currently used in many countries of the world and even at global scale (Vitolo et al., 2020). Other common fire danger indices are the Australian McArthur, the USA NFDRS or the Ketch-Byran index (McArthur, 1967; Deeming et al., 1977; Keetch and Byram, 1968). Most of these indices provide reasonable information on the potential of a wildfire to spread and cause damage. However, these indices do not evaluate the context of the atmosphere around the fire.

Wildfires are phenomena driven by fuel conditions, topography and weather. Fuel conditions are determined by rainfall, temperature and relative humidity among others factors. Some of these factors, such as wind and rainfall, are very dynamic in time while others are static and characterize the local fire behaviour, such as topography. The behaviour of extreme fires, which prevents the evacuation of the affected areas and the possibility of fire extinction, is often related to the interaction of the wildfire dynamics and the conditions of the atmosphere around it (also known as coupled effects). There are multiple factors that define the fire spread behaviour. The relevance of a given factor as the most important of dominant for the spread sometimes define the type of fire. One single fire event may transition from one type to another. For instance, examples of these types are fuel dominated fires or wind driven fires. In this work we will refer to fire driven by convection when convection is an important driving factor of the fire spread.

A first approximation to take into account the atmospheric instability around wildfires was the Haines index (Haines and Service, 1988), which is usually computed between two different heights of the atmosphere above the fire. In cases of extreme wildfire events, the Haines index (Potter, 2018a) can identify dangerous fire spread conditions by using values of temperature and humidity at different elevations. Although the index was successfully used in large fire episodes in the USA (Potter, 2018b), it failed to explain or predict the behaviour of extreme fires that have happened recently (Pinto et al., 2020)).

The main objective of our work is to evaluate the proposed fire danger index (EFBI), which considers deep moist convection, as this is not usually taken into account in most traditional fire danger rating indices applied at global scale.

In some cases, the Haines index saturates (Potter, 2018a), remaining at its maximum value, as in the case of the fire in Pedrógão, in Portugal 2017 (San-Miguel-Ayanz et al., 2019). The aim of the EFBI is to summarize the factors that may imply

a change in fire behaviour and to determine how easy is for the change in fire behaviour to happen. Several authors used factors for storm forecast such as the convective available potential energy (CAPE) and convective inhibition (CIN) to address fire behaviour (Moncrieff and Miller, 1976). Conditioned atmosphere stability was often a common factor of extreme wildfire events. Consequently, increasing the temperature and/or humidity at the surface could lead to atmospheric instability; causing local, dangerous and unexpected conditions.

Wildfire behaviour can be modeled by forest fire simulators, which provide a forecast of the fire intensity and spread. These simulators are based on semi-empirical (Rothermel, 1972) or physical (Mell et al., 2007) models and can be coupled with atmospheric simulation models (WRF-Fire, (Mandel et al., 2011), MESO-NH ForeFire (Baptiste Filippi et al., 2009)) or wind field production models such as Windninja, (Forthofer et al., 2009) to assess wildfire behaviour. However, as the numbers of coupled models grow, the simulation becomes computationally more expensive and it is difficult to gather the required input data without strong uncertainties. Therefore, fire danger ratings and the experience of meteorologists, civil protection officers and fire analysts play an important role in decision making. Moreover, it is not feasible to use coupled models for simulating all wildfires detected at global scale. The capacity to forecast the conditions under which these critical fires can develop is thus of paramount importance and essential for the prevention of damage to the population and human assets. We hereby present a fire danger index referred to as the Extreme Wildfire behaviour Index (EFBI), which looks into the ease of interaction of the wildfire dynamics with the surrounding atmosphere, targeting deep moist convection, and determines if the behaviour of an ignited fire can become critical, allowing it to develop into an extreme wildfire (Duane et al., 2021). The size distribution of the fire events is unbalanced, being more common the fires of small size with an expected fire spread speed. Although less frequent, there is a set of of fire events with high fire spread that overwhelms fire fighting capabilities and reach larger amounts of burnt area. A subset of these fires, considered as extreme fires, interact with the atmosphere around them leading to unexpected or erratic behaviour. Among these extremes fires, some are characterised by the occurrence of pyrocumulus (pyroCu) and pyrocumulonimbus (pyroCb), analysed in Lareau and Clements (2016). The occurrences of these phenomena led to works that analysed the pyrocumulus (Tory et al., 2018), and to the assesment the potential of pyroconvection (Leach and Gibson, 2021; Tory and Kepert, 2021) using different approaches and, for instance, including the moisture and heat released by the fire (Potter, 2005). In the last decades the amount of extreme fires has become very relevant and it has increased awareness about fires as dangerous natural hazard. Besides, the relation of climate change with those events has been studied (Di Virgilio et al., 2019) relying in regional records (McRae et al., 2015) (for Australia) and using well known fire danger indices as McArthur Forest Fire Danger Index(FFDI) (McArthur, 1967) and Haines Index (Haines and Service, 1988). At global scale, the existing records of fire data are quite fragmented and there is much more uncertainty whenn compared to using methods and data suitable for specific regions as mentioned in fire related studies at the global scale as Bowman et al. (2017).

We used a global wildfire database (GlobFire (Artés et al., 2019)) and ERA5 weather reanalysis data (Hersbach, 2016) to compute the EBFI over a set of wildfires. Our approach is suited to be computed at global scale using operational weather forecast models to determine when an ongoing fire can develop extreme wildfire behaviour due to deep moist convection. First, Section 2 describes the proposed index, the workflow for wildfire event selection from the global wildfire database and the

retrieval of the meteorological data and the description of three study cases. The results of the EFBI for the study cases are explained in Section 3. Finally, the conclusions are presented in Section 4.

## 2   Data and methods

The proposed EFBI relies on the premise that the atmosphere can cause a wildfire to be and become convection-driven or that the wildfire can disrupt the atmosphere creating a convective trend. The EFBI is based in well known indicators of convection as 2 and 3, which are also used in Potter (2005) to demonstrate the effects of the increased moisture due to the water released during the fire combustion or the warmed air at ground surface. In addition Lareau and Clements (2016) measured and analysed atmospheric data, using LIDAR and radar, to study pyroCu and pyroCb. Lareau and Clements (2016) showed how the plume condensation level is higher than the ambient lifting condensation level (LCL). Besides, they suggested to use the convective condensation level (CCL) for estimation of the convective temperature for a best representation of the plume condensation level. Also, Tory et al. (2018) proposed a method to estimate the firepower threshold required for pyroCb potential. Leach and Gibson (2021) proposed a parcel-based model to assess how the atmosphere will affect a growing wildfire plume, including heat and moisture released from the fire. In this work we propose a method which computes the current convection trend of the atmosphere using CAPE and CIN, then estimates the temperature required to have a CIN >= 0 in the parcel, also known as convective temperature, which can be obtained computing the CCL. Finally, we compute the convective energy difference between the modified profile and the original one. Most of the past works were based on data gathered on the field, study cases or a database classified as potential pyroCu or pyroCb (as in Di Virgilio et al. (2019)). In this case, we do not estimate the heat and the water release of the fire (as in Leach and Gibson (2021)) but we evaluate the information contribution from a relatively simple approach to estimate moist convection which is feasible to be applied with global weather forecasts. Therefore, we assume that the difference between the surface temperature and the convective temperature is a potential indicator of a change in potential convective energy and can trigger the change in fire behaviour. More physically detailed approaches has been proposed like the Blow Up (BU) $\Delta T$ term in Leach and Gibson (2021) and the $\Delta T_f$ term Tory et al. (2018) has been proposed for pyroCb/Cu assessment or plume dynamics, but they that cannot be analysed at global scale with the available open datasets.

The extreme fire behaviour index(EFBI) determines the amount of increase in temperature degrees at the surface required to cause a null CIN and quantifies the change in the available convective energy. The amount of change of convective energy per degree is used as indicator of ease of deep moist convection.

The EFBI is computed as follows:

$$EFBI = \frac{((CAPEP + CINP) - (CAPE + CIN))}{\Delta T} \tag{1}$$

$$CAPE = \int_{Z_f}^{Zn} g \frac{(T_{v,parcel} - T_{v,env})}{T_{v,env}} dz \tag{2}$$

$$CIN = \int_{Z_{bottom}}^{Z_f} g \frac{(T_{v,parcel} - T_{v,env})}{T_{v,env}} dz \qquad (3)$$

$$\{\Delta T \in \mathbb{R} | (\Delta T = T_{bottomConv} - T_{bottom}) \wedge CIN \geq 0 \wedge \Delta T \geq 0\} \qquad (4)$$

Where CAPE and CIN are defined by Eq. 2 and 3 (Moncrieff and Miller, 1976; Williams and Renno, 1993) and $\Delta T$ is the change in temperature degrees required to achieve a null or negative convective inhibition energy (CIN) (Eq. 4). $\Delta T$ is obtained with the difference between the surface temperature and the increased temperature. The increased temperature is obtained from the CCL as stated in Lareau and Clements (2016) as a more accurate way to estimate the pyroCb/pyroCu initiation heights and implies $CIN >= 0$. CAPE and CIN are recomputed with the increased temperature ($T_{bottom} + \Delta T$) and called CAPEP and CINP, with CINP being zero or greater than zero (no inhibition). $Zn$ is the height of the equilibrium level; $Z_f$ is the height of the level of free convection(LFC); $g$ is the acceleration dur to gravity; $T_{v,parcel}$ is the virtual temperature of the a given parcel; $T_{v,env}$ is the virtual temperature of the environment; $Z_{bottom}$ is the lower height; $T_{bottom}$ is the temperature at the lower altitude; $T_{bottomConv}$ is the temperature at the lower altitude that causes $CIN >= 0$.

In cases in which the atmosphere is already unstable, CIN is equal or greater than 0 being $\Delta T$=0, the values assigned to the index are the full integration of CAPE+CIN.

The EFBI is expressed in J Kg-1ºC-1, which is the amount of energy exchange per unit of mass and per degree of temperature. The value can be used as an estimation of potential wildfire-atmosphere interaction.

High values of EFBI point to a sudden change of energy per mass for a small temperature increase (low value $\Delta T$ and high value of total convective energy). Under these conditions, air can potentially move vertically creating local conditions which are not explicitly provided by meteorological forecasts. When the values are low, the magnitude of the change is small and/or a higher temperature is required at the surface to cause any change. This information is essential for firefighters, since local eddies and sudden weather changes can occur and lead to very fast ember spotting fire spread, which create dangerous and unpredictable conditions for the front line safety (Lareau et al., 2018a).

To test the EFBI, we propose two machine learning approaches, a decision tree and a multilayer perceptron. EFBI and FWI were used to evaluate the discriminatory potential between two fire classes: small fires (500ha during more than one day) and large fires (10000ha in one day). Fire events were extracted from the GlobFire database. The fire size thresholds were chosen to consider both average and extreme fire spread. Small size fires were selected in the same zones where the large fires were selected, but in a different year, always outside the area burnt by the large fires. Using the same zone for both sets of fires we limited the variability in fuel and topography, ensuring that the differences in fire behaviour were due to the meteorological conditions under which the fires evolved. Often small fires are caused by fire spotting or as the result of agricultural practices. To avoid these types of fires, it was required that the fires had more than 90% of the burnt area in wild land vegetation. The final selection was a total of 445 cases, with 223 fires larger than 10000 ha and 222 fires smaller than 500ha. Figure 1 shows the distribution and the year of each fire at global scale.

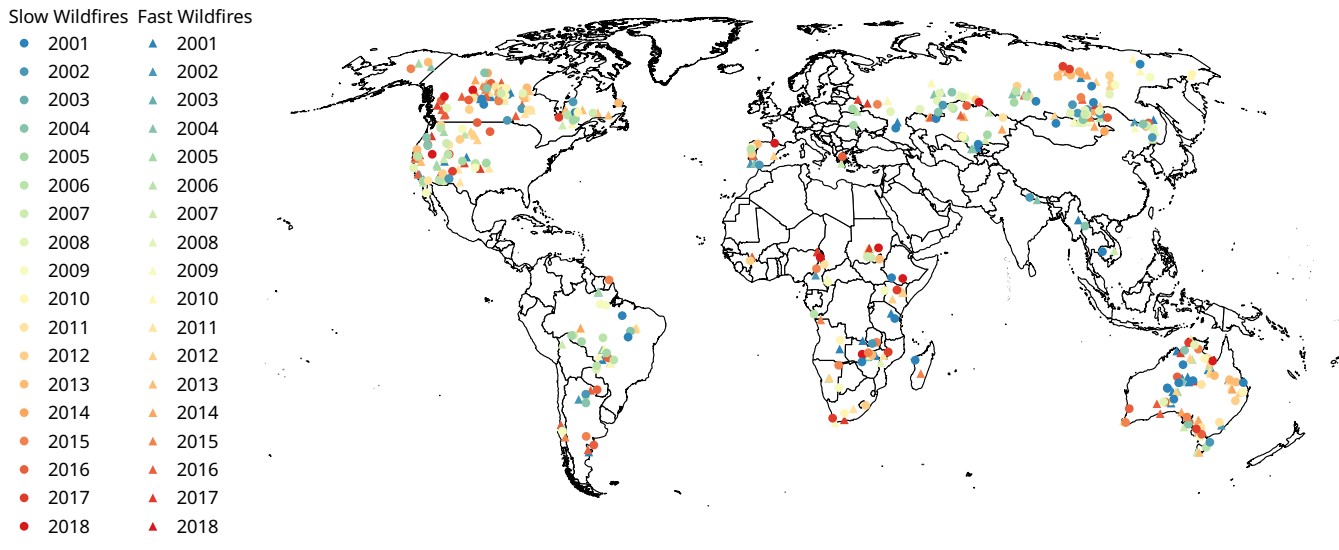

**Figure 1.** Location of the wildfires coloured by year. Fast fires and slow fires are depicted with triangles and circles respectively.

For each fire, the Canadian Fire Weather index (FWI) and all its components were retrieved from the ECMWF ERA5 dataset at 0.25 degrees resolution (Vitolo et al., 2020). In order to differentiate between agricultural fires or wildfires, the land cover from the Climate Change Initiative, CCI, (Defourny et al., 2012) was used, requiring that 90% of the burnt area was forest or shrubs, and thus avoiding fires in crops or agricultural areas. For each selected fire, temperature, relative humidity and wind

profile were retrieved for all the heights above the wildfire event using the application user interface (API) of the Climate Data Store (CDS) for the ERA5 reanalysis dataset(Hersbach, 2016). Then, the EFBI and the vertical profiles for every event were computed, producing skew-t plots for every time step and generating plots with the time evolution of the index and the factors used for the computation of the EFBI. The aim of the skew-t plots is to visualize the ease with which the fire may become convection driven given an atmosphere context.

All the parameters obtained from FWI and EFBI were used to compute the mutual information(MI) of the wildfires that spread more than 10000ha in one day(large) or less than 500ha in more than one day(small). The MI has been chosen to quantify the individual importance of the variables gathered for each fire (such as the FWI components and EFBI) in relation to the speed of the spread of the fire, which is categorized into either small or large fire events. The FWI was computed for an entire day, and EFBI was computed with one hour time steps. Therefore, the values of the EFBI were aggregated using the

minimum, the maximum and the average value for the time window. In addition, the date associated with the burnt area was based on GlobFire which uses the burnt area product of MODIS MCD64A1(Giglio et al., 2015). Due to clouds or dense smoke plumes, it is possible that a fire might have not been detected until some days after its ignition. To account for this delay, the initial day of the time window for the FWI and EFBI was decreased by two days. The maximum value of the FWI and its

components during the time window were selected. When applying FWI at global scale its values are not comparable between different locations. Despite that fact, the percentiles of FWI can provide information about the fire danger for a given area. Therefore, the values of the FWI were used in percentile values, computed for the period 1979 to 2019 of the same week of the year.

In addition to the proposed EFBI and FWI, CHaines index is also computed using the same ERA5 data used for the EFBI. For the temperature depression term, we used the temperature at 850 and 700 hPa. For the dew point depression term, we used temperature at 850 hPa and dew point temperature at 850 hPa.

The MI for continuous values was computed using the method described in (Kraskov et al., 2004), (Ross, 2014) and using (Pedregosa et al., 2011). These methods use a nearest neighbours approach with a random initialization; for that reason, 10 additional attributes were added to the analysis with random values to evaluate the noise of the MI. The MI computation was done 1000 times. The computations of MI were initialized with random values to evaluate the noise of the attributes for the MI computation.

In addition, the behaviour of the EFBI was shown analysed fr three different study cases. First, we analysed the fire in Pedrógão Grande, Portugal in 2017 using high spatial and temporal resolution fire perimeters. Next, the EFBI was computed with forecast data using for the analysis the daily fire perimeters of the wildfire that took place in Robore Bolivia in 2019. Lastly, we studied the spatial distribution of EFBI for the set of extreme wildfires in the southeast coast in Australia at the end of 2019.

## 3   Results

EFBI results are evaluated combining the resulting values of the EFBI and FWI (Vitolo et al., 2020) to predict extreme fire behaviour observed in GlobFire using weather data from ERA5 (Hersbach, 2016). In this work, a machine learning approach is used to check the feasibility of using the EFBI to predict extreme wildfire behaviour at global scale. In addition, the EFBI values are shown for several study cases.

In Fig. 2, we illustrate how EFBI components behave for a given vertical profile at given time step. The blue area is the convective inhibition (CIN), and the red area (from level of free convection (LFC) point to equilibrium level (EL)) is the convective available energy (CAPE). When CIN values are low and CAPE values are high, wildfires can be convection driven at high altitudes. A temperature increase at the surface can reduce CIN values and increase CAPE values. Figure 2 shows a Skew-t plot with the parameters used to compute the EFBI. The figure shows the vertical profile of the atmosphere with the temperature as a red line and the temperature dew point in green at different altitudes(right vertical axis). These parameters are enough to compute the amount of work caused by the buoyancy force. By modifying temperature and dew point temperature the buoyancy changes, and consequently, the trend of the air to move. Figure 2 shows how a change of 14 degrees Celsius at the surface temperature causes a considerable change in CAPE and CIN values with a CAPEP close to 2443 Jkg-1 and a low value of CINP. The temperature increase, 14 degrees Celsius, is only used and indicator of moist convection trend by an increase of heat at the surface, since the biomass burnt, heat and gases released by the fire are not estimated the heat dissipation is not

simulated or estimated (Byram (1954), Morvan and Frangieh (2018) and Tory and Kepert (2021)). Therefore, the amount of work due to the buoyancy change, passes from 34 Jkg-1 (CAPE) to 2443 Jkg-1(CAPEP) when CIN is not an inhibition, i.e. passing from a negative value to a positive value. This leads to a total change of 233.531 Jkg-1ºC-1 per each degree increase at
the surface. The CAPEP is around 2443 Jkg-1 when thunderstorms often exceed CAPE values of 1000 Jkg-1 and 5000 Jkg-1 in extreme cases (Lombardo and Colle, 2011). The increased temperature $\Delta T$ can not be understood as a real temperature increase but as an indicator, since it would not be realistic that an entire parcel receives enough energy to increase the surface temperature instantly being isolated from the neighbouring parcels and upper layers. In this work, we assume that EFBI can be an indicator of ease of deep moist convection but we do not take into account all complex factors involved in fire behaviour and
atmosphere interactions(Sullivan (2017)). Figure 2 also illustrates how the CCL is used as suggested by Lareau and Clements (2016) as a better estimation of the convective temperature for a best representation of the plume condensation level. From the CCL temperature, following the dry adiabatic up to the surface, we obtain the temperature which would neutralize CIN and increase considerably CAPE.

It is worth mentioning that CIN values at lower heights are also related with the temperature dew point at the surface (green
line at 1000hPa). With a constant temperature at the surface and an increase of dew point temperature, the CIN is reduced and therefore the EFBI increases. This may happen with a sudden increase of humidity at the surface, without any increase of temperature (already taken into account in the computation of the EFBI).

The timeline of the EFBI for the wildfire in Sala (2014, Sweden) is shown in Fig. 3. The time period when the fire had an extreme behaviour, as documented by firefighters on the ground, is delimited with vertical dashed lines. In this period, the
modified convective available energy reaches values above 2500J*kg-1 and the index values are near 800 Jkg-1ºC-1.

The EFBI is computed for each time step, as explained in the previous illustrative example, for all the cases extracted from GlobFire database using the method described in section 2. Regarding the information of the different features used to discriminate small from large fires, Fig.4 shows the values of the MI mean and standard deviations of each attribute for the 1000 iterations. The use of only the minimum, maximum or average of the EFBI provided more information to separate small
from large fires than the percentile and value of the drought code (DC) of the FWI. Figure 4 does not show the discriminatory power of all variables, but the amount of information with which each of the factors individually contribute to the classification between the two classes. EFBI on its own cannot easily discriminate between the classes without using factors included in the computation of the FWI such as fine fuel moisture content (FFMC), duff moisture code(DMC) or drought code(DC). The discriminatory potential between the two classes using the EFBI is very low without using the FWI. Regarding the MI
of CHaines index, the minimum value provides a considerable amount of information, more than the minimum value of the EFBI. However, the maximum value of the CHaines index provides a very low amount of information when compared to the maximum value of the EFBI, which provides valuable information for the classification. That fact could point out to a potential saturation of the CHaines index, meaning that the maximum value is not giving information to know if the fire is small or large.

For instance, the EFBI can have high values after a rainfall event. It is thus important to stress that the EFBI becomes
relevant when combined with the FWI components. A decision tree based on information gain as criteria was used to show the relevance of the EFBI in discriminating small fires from large fires. Figure 5 shows the decision tree, which is based on entropy

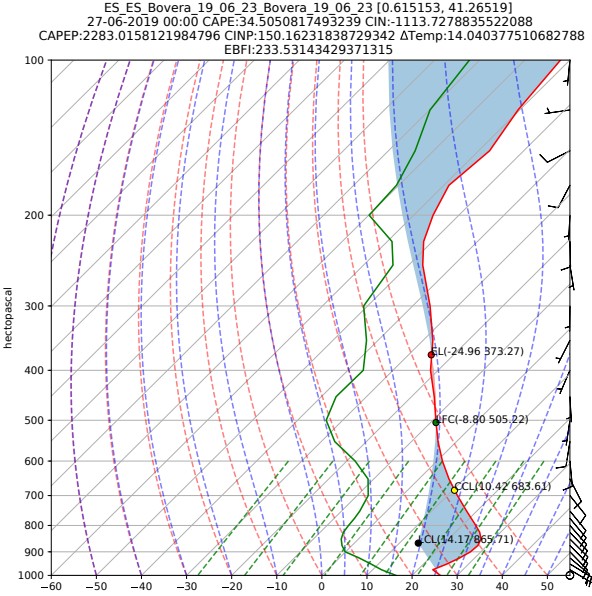

**Figure 2.** Sample of one Skew-t plot for a given time step for the wildfire that took place in Torre del Español (Spain) that burnt 6625ha from 26th to 27th June of 2019.

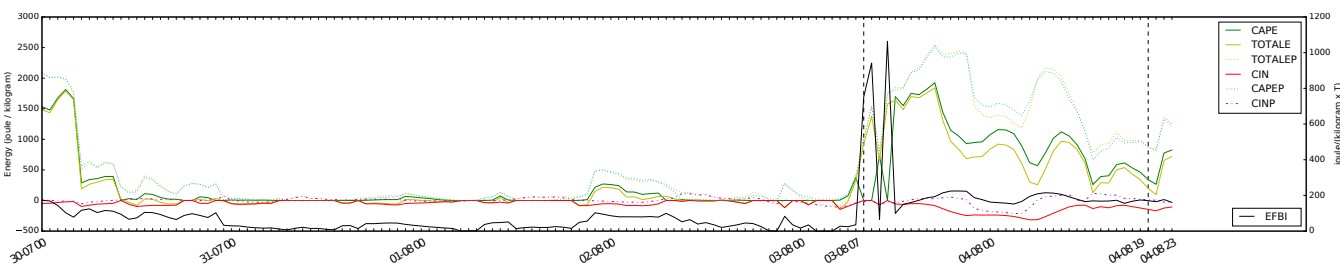

**Figure 3.** Sample of time line of the EFBI (right vertical axis) and the factors used (left vertical axis) for the wildfire in Sala (2014, Sweden) that burnt close to 10000ha from 3h to 4th August of 2014 (time period marked with vertical dashed lines). Horizontal axis is the time line using hourly steps in format dd/mm hh.

reduction with a maximum depth of 4 levels (for visualizing purposes). The root uses the daily severity rating (DSR), which is a transformation of the FWI. At the second level, the maximum value of the EFBI is used, before considering any of the FWI components such as the percentiles of the initial spread index (ISI), the drought code (DC), the built up index (BUI) and the fine fuel moisture content (FFMC). Therefore, EFBI helps to reduce the entropy when discriminating between the two classes at global scale, being more relevant than the ISI or DC percentile.

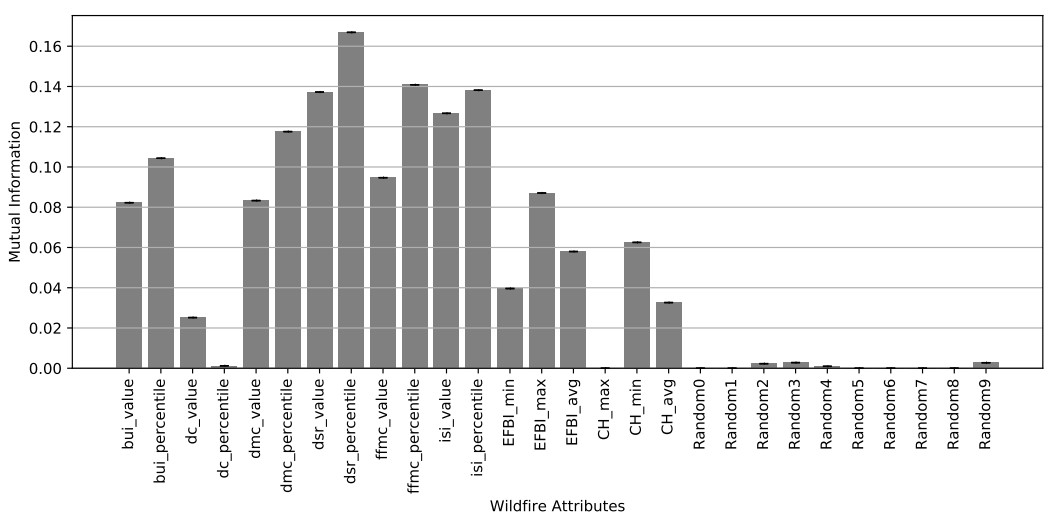

**Figure 4.** Mutual information of the different attributes gathered for each fire regarding their tag as fast or slow fires.

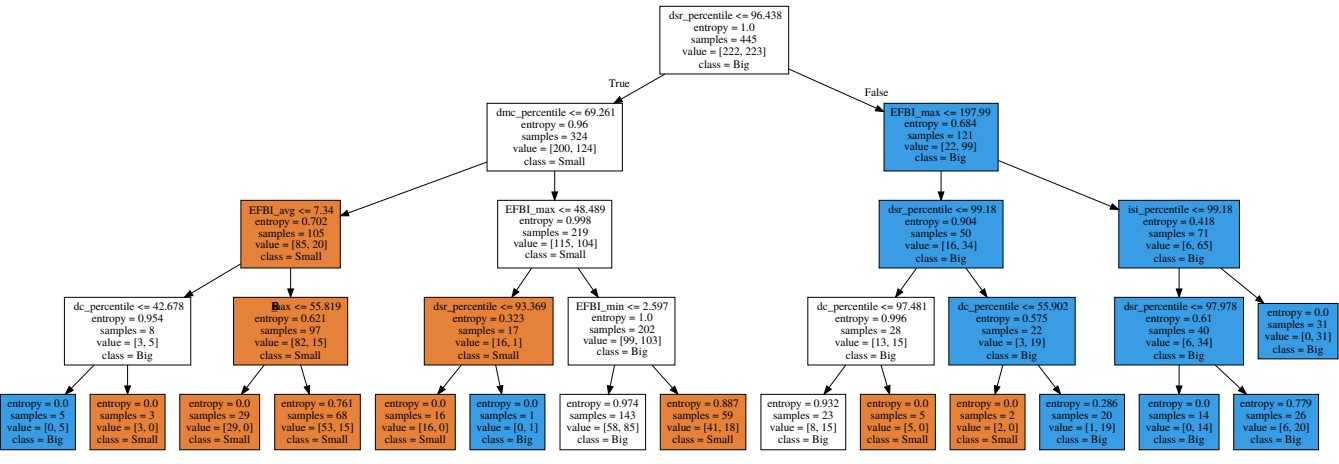

**Figure 5.** Decision tree based on entropy reduction build with the 445 wildfire events (222 small and 223 large wildfires).

A machine learning approach was applied using a decision tree and a multilayer perceptron. Removing the maximum depth restriction to the decision tree and performing a cross-validation of 1000 splits with 5% of cases, as a test, the average accuracy was 65,13% with a standard error of 7.29%. The accuracy was computed using Scikit Learn considering the accuracy of the percentage of each sample that each label is correctly predicted.

Applying a multilayer perceptron with 3 hidden layers of 300 neurons each with ReLU activation and using a Adam solver, (KingaD, 2015) optimizer for training, and the same parameters for a cross-validation, an accuracy of 78,37% (standard error 1.85%) was reached. If the parameters related with the FWI were exclusively used, the accuracy decreased to 60.75% (standard

error 2.24%). Using FWI with CHaines raised the accuracy to 63.42% (standard error 4.12%). All input values have been normalised for the three cases (FWI, FWI+EFBI and FWI+CHaines).

As previously mentioned in Section 2, the dataset used to test the EFBI was obtained using a fully automated process. The information regarding the fire behaviour was obtained from GlobFire using SQL queries. The query guarantees that there are no fires happening in a 2 degree radius around a detected fire during a time period of 30 days. During the cross-validation process, the wildfires that were more frequently misclassified can be identified. When looking at the first 50 wildfires that are often misclassified, the proportion of the cases is very balanced between true small and large fire cases. Assuming that such cases are a source of noise and can be deleted from the dataset, the accuracy rises to 80.30% (standard error 1.94%). It should be noted that some cases could be false positives or false negatives. False positives could happen when a fire was burning in a cloudy area for several days. Once the area is free of clouds, the burnt area is detected for that day, even it was burning for days before this day. False negatives may have happened with re-ignitions of large fires. Besides, large fires are not always convection driven or related to an unstable atmosphere. The current automated process to discriminate potential large fires using the GlobFire database can be implemented in near-real time, using weather forecast data, allowing the potential identification of dangerous convection driven fires in advance, and thus increasing wildfire danger rating. This, in turn, would increase preparedness for firefighting procedures and enhance the safety of crews on the ground. Additionally, the overall procedure in the discrimination of potential large fires can be improved if the dataset is manually checked and the fire types are better defined.

## 3.1 Study Cases

### 3.1.1 Pedrógão Grande, Portugal 2017

This wildfire had one of the most severe fire behaviour in Europe. The fire was ignited on 17th June and ran until 23th June. Figure 6 shows an explosive expansion from 17th June to 18th June, which is followed by a constant but severe fire expansion. For this analysis the fire perimeters made for a wildfire report done by a technical commission (Guerreiro, 2017) were used as reference for the fire spread.

From 17th to 18th June there was an increase of burnt area larger than 20000ha in one day.

Figure 7 shows the value of the EFBI and its components during the entire month for the fires computed from the ERA5 reanalysis; the two vertical dashed lines in the figure delimit the duration of the fire. The EFBI shows that there is a considerable potential for the interaction of the fire with the atmosphere. In addition, during the days of the fire there was a natural CAPE of nearly 5000J*kg-1 inhibited by a small value of CIN. Increasing the temperature at the surface removing the inhibition, would produce a sudden convection. The index values are close to 250 J*kg-1 per temperature degree increased and a total convective energy about 6000 J*kg-1.

This case shows a natural tendency towards a convection driven behaviour that may be caused by the atmospheric instability itself, without the need for a considerable amount of heat and/or an increase in relative humidity at the surface.

For this case study, detailed fire perimeters for each time step were available, which made it possible to analyse the relation between the maximum fire spread and the EFBI. Figure 8 shows a scatter plot where each point is a time step given an EFBI

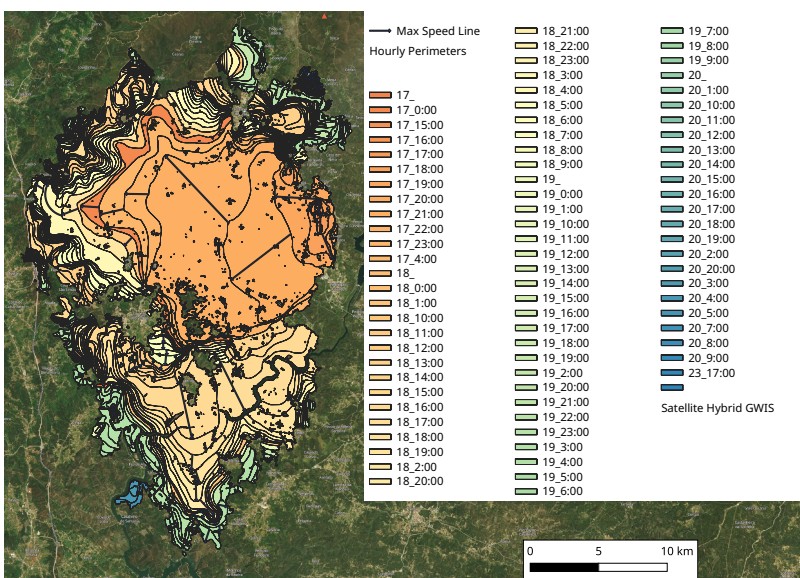

**Figure 6.** Time sequence of the wildfire which took place in Pedrógão Grande (Guerreiro, 2017), Portugal on 17th June of 2017. The maximum speed line between time steps is shown with a black arrow. Background image ©MapTiler (https://maptiler.com/copyright).

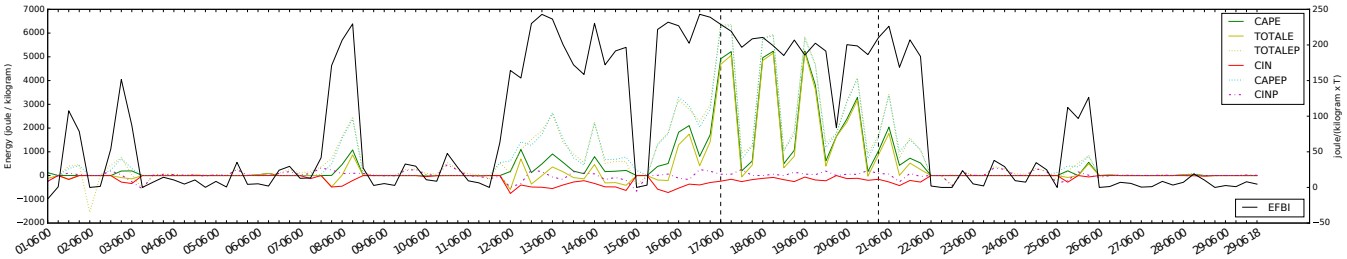

**Figure 7.** EFBI and its components for the wildfire in Pedrógão on June 2017 for the fire centroid (-8.2252, 39.952).

value and the estimated maximum fire speed. While a wide variety of values of EFBI are shown for low fire spread speed values, considerably higher EFBI values are shown for high speed values, compared with the rest of the point cloud. However, there is a weak correlation between the speed and the EFBI. It is worth mentioning that the blow up of the wildfire took place at the beginning of the event; assuming that the fire was in convection almost from the first time steps, EFBI and fire spread speed may not show a strong correlation when looking at all the hourly time steps. However, Fig.8 shows that, even with an ongoing convection driven fire, the atmosphere stability context computed from a numeric model is still important and speeds faster 1 km/h are only present when EFBI is above 220.

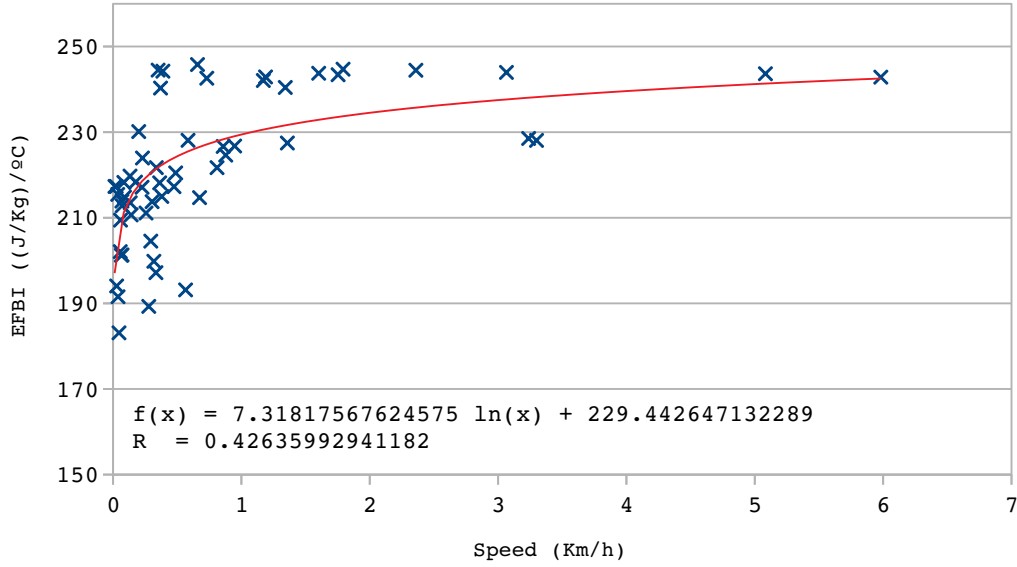

**Figure 8.** Scatter plot showing the maximum fire front speed and the values of the EFBI for each time step with a logarithmic trend line.

### 3.1.2 Forecast use Roboré, Bolivia 2019

The values of the EFBI were computed using ERA5 forecast for the fire that took place in Roboré, Bolivia in 2019. This wildfire lasted 2 months and had a very variable fire behaviour. Figure 9 shows the daily burnt area for this wildfire using the GlobFire database. However, the level of detail of the fire mapping for this study case is not as accurate as in the previous case shown in Fig. 6.

For this case, the EFBI was computed with three different deterministic forecasts of ERA5 on August 15 and 29, and
September 3. During this wildfire, the EFBI forecasted different periods of extreme behaviour caused by moist convection, although the correlation was not as clear as in the case shown in Section 3.1.1. For long duration wildfires that interact with the atmosphere it can be assumed that once a wildfire interacts strongly with the atmosphere, the trends of the EFBI are no longer valid, since they are based on a forecast that does not include the interaction. However, in the above case of Bolivia, our results show that, despite the fire-atmosphere interactions, the overall moist convective trend may remain important and the values of
the EFBI can be relevant for a second convection interaction. It is not feasible to find a correlation between the fire spread and the EFBI values in this case because the quality of the fire mapping is daily, with a potential uncertainty of days of the maps. Also, the weather forecast that has been computed previous to a potential wildfire-atmosphere interaction may not depict the atmosphere around the fire just after a wildfire-atmosphere interaction.

Figure 10 shows the computation of the daily EFBI average joining the data from the different forecasts. In addition, the
GlobFire database was retrieved and used to estimate the maximum daily fire run. Since the daily burnt area could have been mapped with some days of delay, an average was applied to the maximum daily fire run using a time window of the 2 previous

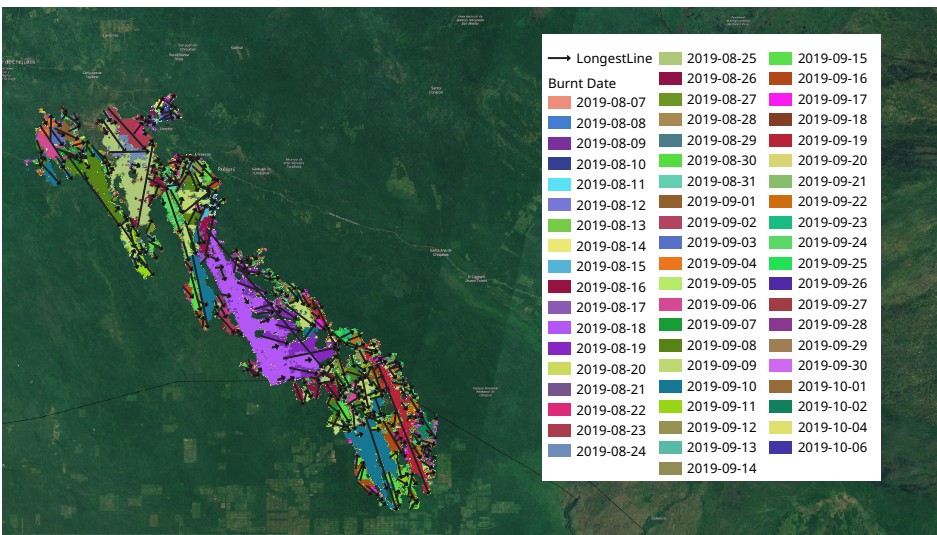

**Figure 9.** Daily burnt area for the duration of wildfire in Roboré, Bolivia in 2019. The maximum speed line between time steps is shown with a black arrow. The missing days do not have any daily burnt area in GlobFire. Background image ©MapTiler (https://maptiler.com/copyright).

days. Figure 10 shows a peak of the EFBI followed by another one which started on 16th of August and a third peak on 22nd of August. After the first peak, the maximum fire longitude of the daily burnt areas had increased from the 16th to the 20th of August, while the highest runs of the fire happened between the 18th and 20th; the fire activity has another peak after
August 22 having two observed pyroCb on 18th and 25th of August. After the 25th August, the EFBI trends seem to be totally uncorrelated with the fire runs until 1st September. Afterwards, there is another peak on 7th of September which also seems to affect the fire runs, with 2 observed pyroCb on 7th and 8th of September. Later, EFBI and the maximum line length are again uncorrelated until 17th September when there is again a relation at the last peak of the run, when another pyroCb was observed.

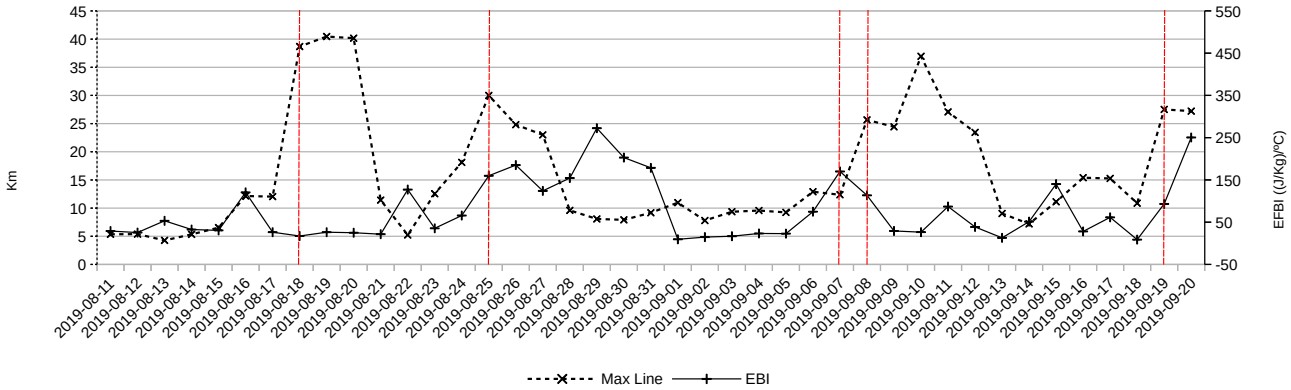

**Figure 10.** Maximum longitude of the the daily burnt area and the EFBI and its components for the wildfire in Roboré, Bolivia 2019. Vertical dashed red lines show when a pyroCb took place.

An example of pyroCb seen with Sentinel 2 displayed in Fig. 11, where parts of the plume are shown in white.

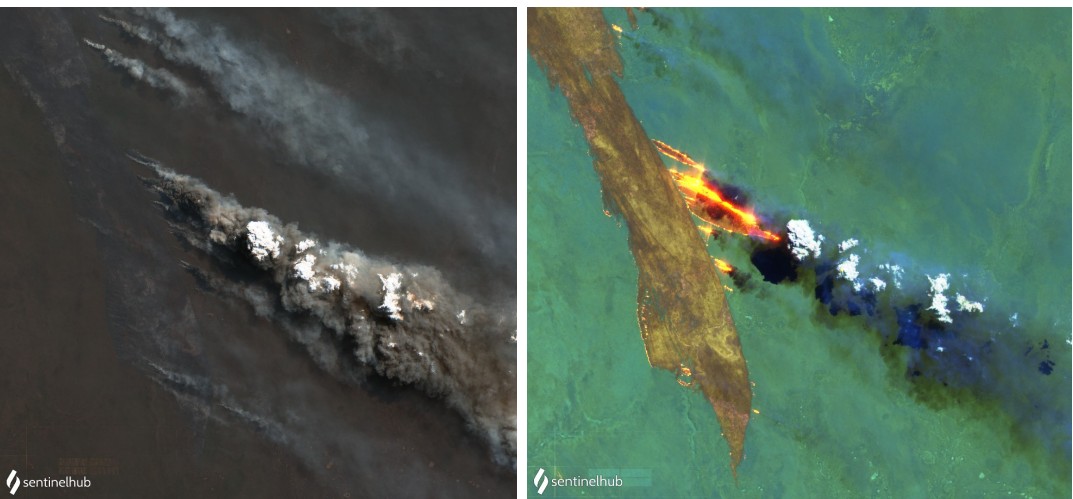

**Figure 11.** Sentinel 2, natural colour and atmospheric penetration, from right to left from S2 on 18th August. Roboré, Bolivia.

### 3.1.3 Wildfires Australia 2019

The EFBI was computed with ERA5 reanalysis for this area at the time of the fires showing very high values. Figure 12 shows the maximum values of the EFBI overlapped with the active fires (thermal anomalies shown as black points) that took place the next day.

It is worth mentioning that the assimilation system used to generate the ERA5 reanalysis dataset could partially affect the behaviour of the EFBI. In reanalysis, some of the consequences of heat release from the fire could be taken into account, while in the forecast for a given day those conditions would not be considered.

## 4 Conclusions

This work demonstrates that simple metrics of the atmospheric stability could provide valuable information for enhanced fire danger rating applied at global scale, increasing preparedness and improving safety and efficiency of firefighting. The EFBI could be used to detect days in which fires could exhibit extreme behaviour on a global scale. On those days, fires could add an unpredictable interaction with atmospheric vertical profile, increasing fire behaviour due to moist convection. However, EFBI is targeting only deep moist convection and is subject to several factors which could cause uncertainty due to the wildfire-atmosphere interaction during a forecast and to the resolution of data used. Also, convection driven fires are not the only kind of fires that can exhibit a extreme fast spreading. Despite this, the approach used to differentiate between fast and slow spreading fires at global scale demonstrates a potential use of the EFBI combined with the FWI to forecast extreme fire

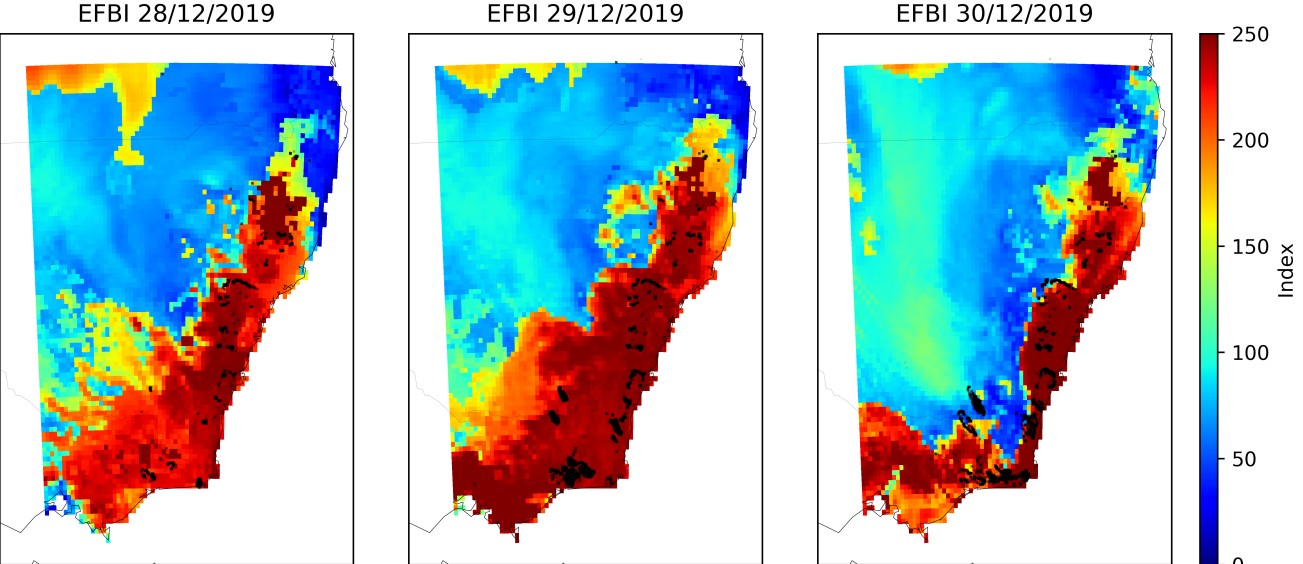

**Figure 12.** EFBI from 28/12/2019 to 30/12/2019 overlapped with the active fires (black dots) that took place the next day over southeast Australia.

behaviour. Moreover, the data used for the vertical profiles could be improved by estimating the surface temperature, relative humidity at the surface using the altitude of the fire event and evaluating the air dryness at different levels.

The EFBI holds the potential to improve fire danger forecast ratings applied at global scale. The initial testing presented in this article reached a 78% accuracy, discriminating the final size of fires between small($<=$500ha) or large ($>=$10000) at global scale. Accuracy could be improved by increasing the number of classes and using other input factors to discriminate wind driven fires in small bushes or grass which can become large fires but have low probability of becoming convection driven wildfires or breaking the atmosphere stability. These big fires are not discarded in this work. For instance, dry and strong catabatic wind-driven fires like those of the Mistral (Western Europe), Santa Ana and Diablo winds (California) or Zonda wind (Argentina and Chile) are not always fast wildfires caused by convection. This kind of fires are not convection driven but are extreme fires anyway and could reduce the accuracy reached in this work. This might have influenced the study case in Roboré (Bolivia), which generated several pyroCb but had also wind driven fire spreads.

The EFBI can be computed at least twice per day at global scale with up to 10 days forecast. For instance, using the Global Forecast System(GFS) for a 10 day forecast at 0.25 degrees of spatial resolution (approximately 25 km), the computation of the EFBI at the global scale took 4 hours in a single node with linear speed up, using multiple cores.

The EFBI has shown high discriminatory power of large fires using ERA5 at 0.25 degrees resolution at hourly steps and GlobFire at 500m with a temporal resolution of 1 day. It showed a considerable relevance in the mutual information and in the discriminative power with the decision tree. It should be noted that the training of the neural network with only FWI values and its components, which are the fire danger indices most used worldwide, dropped the accuracy from 65.5% to 58%. Our

results also highlight the relevance of the GlobFire dataset to analyse fire behaviour and improve current danger ratings for extreme wildfires that may happen more frequently worldwide under climate change. This work also stress the importance of developing datasets for fire behaviour improving temporal and spatial resolution. Such datasets would enable to apply and validate complex models as those proposed by Tory et al. (2018); Tory and Kepert (2021) and use detailed analysis as done, for instance, by Lareau et al. (2018a); Bagley and Clements (2021).

Using a high spatial and temporal resolution dataset in the study case of Pedrógão Grande in Portugal, the EFBI showed a close relation with the speed of fire spread. Moreover, using the daily data of GlobFire for the fire behaviour in the case of Roboré, EFBI could have forecasted the convective periods during the wildfire. For the last study case, which includes a cluster of extreme wildfires in Australia, high values of the EFBI match spatially with the occurrences of the such events, although the behaviour of the individual fires within the cluster was not analysed.

The above results indicate, that the EFBI shows a potential to improve the current fire danger rating at global scale by establishing a fire typology, which could characterize potential explosive behaviour of wildfires using deep moist convection estimation and ERA5.

*Author contributions.* T.A. conceived and designed the experiments; J.SMA. directed the research project. T.A. performed the experiments and the creation of the method. T.A., M.C. analysed the data, the method and the overall application. The first manuscript draft was written by T.A. All authors discussed the results and contributed to the manuscript editing process. T.HD provided language check and proofreading.

*Competing interests.* All authors declare that no competing interests are present

*Acknowledgements.* A non-published work between T.A and Thomas Petroliagkis in 2015 led to the proposed method after the development of GlobFire, the customization of Metpy and the publication of ERA5. Reconstructed wildfire spread data for Pedrogão Grande was supplied by Nuno Guiomar and Paulo Fernandes.

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
