# Peer review of "Wildfire-Atmosphere Interaction Index for Extreme Fire behaviour"

_Natural Hazards and Earth System Sciences, 2021_

## Author Response (AR1)

**Reviewer 1: Dr. Miguel Pinto comment preprint NHESS-2021-122**

**31 May 2021**

**In bold anonymous reviewer**

The authors appreciate the comment provided by Dr. Miguel Pinto. We consider the comment s an accurate review of the paper. Then, we try to answer most of the questions and suggestions raised by Dr. Miguel Pinto.

**Reviewer 1: The article describes a new index for improving the extreme fire behaviour predictability by considering the vertical atmospheric profile, namely, to evaluate if conditions are favourable to convective fire behaviour. In general, the article is well written, the methodology is interesting, and the topic is relevant for the fire research community and for practical applications. I have however some concerns and comments regarding the current version of the manuscript.**

**My main concern is that a discussion comparing the proposed index with existing works in the literature is lacking. In the introduction, the authors mention the Haines Index but quickly dismiss its usefulness citing Pinto et al. 2020 and the saturation problem of the Haines Index. The cited work by Pinto et al. 2020 does however address the same problem, proposing an enhanced Fire Weather Index that combines the FWI with the "Continuous Haines Index". I was expecting to see some discussion or comparison to put the proposed EFBI in the context of those existing indices. For instance, since the EFBI uses the vertical profile of the atmosphere, I would assume that the authors have all the data necessary to compute the Continuous Haines Index (that is based on the temperature and dew point at two pressure levels) and the enhanced FWI. I believe such a comparison would be of great interest to the fire community and would set a new validation standard for future works on this topic.**

Authors: In this paper we propose a methodology for an index that intends to cover and improve the concept developed by Haines by also considering conditioned stability, which may change in an ongoing fire. We added the Continuous Haines index in the machine learning part of the paper. As the reviewer suggests, it is interesting to compare the existing indexes.

We added the CHaines in the manuscript; however, an extensive comparison using enhanced FWI, the Chaines at different altitudes, plus the Haines and FWI computed from ECMWF extreme forecast would be an entire new work. We also consider that, for a comparison using machine learning, hyperparameter tuning should be required for each model. The Continuous Haines Index can be computed at different heights. We used it in the Pedrogao fire in 2017, when we computed the CHaines and the Haines indices from the ERA Interim dataset.

The 4 different versions of the CHaines index were:

[Figure]

Figure 6 Continuous Haines values for several pressure levels.

EFBI:

[Figure]

**Reviewer 1: My second concern is that, to my understanding, the EFBI was computed with one hour time steps and aggregated using the minimum, the maximum and the average and the three variables were used as predictors when applying the decision tree and the multilayer perceptron. Since the FWI is computed only daily it is possible that part of the improvement in the accuracy is due to the indirect incorporation of sub-daily conditions. It would be relevant to compare the change in accuracy when considering the minimum, maximum and average aggregates of EFBI individually before including the three aggregates all together.**

Author: As the reviewer comments, the use of EFBI required an aggregation of the data, which was required to be used with the FWI, but also necessary for the use of a relatively simple and common approach with machine learning for both indexes. The sub-daily conditions may become a disadvantage too for the EFBI, since FWI was developed as a daily risk, while the EFBI is a "current" condition index, with a lot of variability which may show much better results when having high frequency data as in the use case of Pedrogao. A set of cases with high spatial and temporal accuracy in the data would allow such analysis, but we wanted to evaluate the information contribution of the index for a set of cases. Although

the use of databases such as GlobFire or FireAtlas was a clear option to find big fires in different conditions and locations, those databases have limitations and it was necessary to add use cases where we could use the EFBI.

EFBI gives information about convection, but not on hourly conditions of wind speed or fuel moisture. In that sense, EFBI is absolutely blind to drought conditions and does not provide information about gusty winds or turbulences. The EFBI has units, it is not adimensional and only provides the amount of energy required for by a parcel of air to change for each degree of temperature changed at the surface. When aggregated by day, EFBI does not contribute with information about the soil moisture, fuel moisture, wind speed or not even the average value of the temperature during the day. The index could be high in very cold areas too. It becomes useful when combined with FWI. EBFI, alone, is not accurate enough to discriminate between large and small fires.

**Reviewer 1: Other comments**

**L20-21: I suggest rewording this sentence.**

Done.

**L116-117: Are the 222 small fires a subset of a larger initial selection matching the described criteria? I would expect the number of small fires to be higher.**

Author: We decided to create a subset balanced and in close areas to the large fires but different years to avoid including small spotting fires around a big fire and evaluate a close location under different conditions, since neural networks are less sensitive to unbalanced number of elements in each class. In the analytical process, what is considered "ground truth" is not such, but the outcome from an automatic selection from a dataset with implicit uncertainty.

**L136: If the initial day of the time window is increased by two days wouldn't the correct day be missed for the cases where the MODIS MCD64A1 gives the correct or the following day? How is the 2-day value selected?**

Author: MCD64A1 provides a lot of very valuable information but includes uncertainties, specially as regards the dates of fire occurrence, which could add noise to the dataset used in the analysis. We increased the time window to avoid large convective fires that may be mapped days after they happened. Then, we evaluated the min, max and average during the time window values for that fire event, so that the right value would be included in the time window. However, as the reviewer mentions, by increasing the time window, we could add some noise to the EFBI min,max values for those fires that are correctly mapped in MCD64A1. Our approach focused on including the large convective fires that are sometimes mapped with some days of delay due to the presence of clouds and plumes (case of Pedrogao with MCD64A1)

**L141-142: Consider updating to: "using Scikit-learn (Pedregosa et al., 2011)."**

Author: We used the citation recommended by the authors of scikit-learn (https://scikitlearn.org/stable/about.html)

**L179-180: I suggest updating to: "than the percentile and value of drought code". In fact, looking at Figure 4, the MI for dc_percentile is not significantly higher than zero.**

Author: Done.

**Figure 5: The EFBI is named as "Index" in the figure. Consider renaming or updating the figure description to make the interpretation clear. The same comment applies to some of the other figures.**

Author: Done, changed in all figures.

**L197: What activation function was used in the multilayer perceptron?**

Author: ReLU. Also solver changed to Adam and only 3 layers with 300 neurons each. Added to the text.

**L198-199: What is the standard error in these cases?**

Author: Added to the text of the article.

**L199: I assume FWI is referring to the set of FWI components in percentile form, please clarify if this is the case.**

Author: Yes, we are using the FWI and components in percentiles. Because except of the DC, gives more information.

**L203-205: It is certainly expected that by removing the 50 most often misclassified cases, out of a total of 445 cases, the accuracy would rise substantially. Unless there is some manual checking of these 50 events, I don't see the point of this exercise.**

Author: The events removed are not only removed from the dataset used for validation. The values are removed for the entire process included the training. Repeating the process with random data should not show any significative improvement, although it does in this case when removing the most misclassified cases. The reviewer is right regarding the fact that cases should not be removed, but instead included in a new class or typology; this is foreseen in future work. Those cases may include fires that are not properly depicted by the data of EBFI and the FWI, or it might be that there is no right class for them with the data used in our analysis. However, our approach is applied at a global scale and using only burnt area products based on remote sensing and weather reanalysis to demonstrate that the EFBI provides information to characterize fires. It would be possible, instead of classifying fires into big or small, to try to classify fires by typology and speed which may be even more complex at global scale and would require, nowadays, manual classification and verification. The exercise only shows that is not random data; these cases are misclassified and this could

be due to the fact they may potentially be fire types which cannot be classified using only two classes, like large or small, using FWI and EBFI.

**L231-232: The results of the case study are interesting; I would further comment that speeds greater than about 1 km/h are only present for EFBI values above ~220. This result is close to the threshold of about 200 in Figure 5 for "Index_max".**

Author: Comment added. Done.

**L270: I suggest adding some comment regarding the need for future research towards constructing datasets of fire behaviour type and higher temporal resolution fire progression.**

Author: Comment added. Done.

**L274: The EFBI is misspelled as "EBI".**

Author: Fixed.

**L281: "ha" is missing after 10000.**

Author: Fixed.

**L291-292: Does the 4-hour computation time considers the time to download the GFS data?**

Author: Yes. Computation is done simultaneously to the download (pipeline approach) except for the first step using a single node of 16 cores.

**Reviewer 2: Anonymous comment preprint NHESS-2021-122**

**12 Aug 2021**

**In bold anonymous reviewer**

We appreciate the work done by the reviewer and the concerns he/she expressed. We would like to remark that due to the previous comments of Dr. M. Pinto we modified the manuscript by including another index such as CHaines and changed the analysis of the neural network, which improved the accuracy of the classification. That modifications are detailed in the answer to Dr. M. Pinto's comment.

**Reviewer 2:"An index designed to identify atmospheric instability that promotes or facilitates extreme fire behaviour (EFBI) is introduced. Indices constructed from surface-based parameters often fail to discriminate between extreme and non-extreme fire behaviour. Decades ago, atmospheric stability was proposed as a missing ingredient in such indices. The EFBI provides a measure of atmospheric stability designed to fill this gap, and be used in conjunction with existing surface-based indices to provide a more comprehensive assessment of fire danger.**

**Results are presented that suggest using the EFBI in conjunction with a surface-based index (the Canadian Fire Weather Index, FWI) does improve the predictive performance compared to only using the FWI.**

**Case studies are included that illustrate the EFBI for specific extreme fire-behaviour events, including pyrocumulonimbus (pyroCb) cases.**

**I have a number of major concerns with this paper:**

1. **It was not sufficiently polished for submission. (It could have benefitted from an internal review before submission.)**

Authors: We assume that most of the points to be polished are detailed in the points of concern expressed by the reviewer, which have been considered in the revised version of the manuscript.

2. **It lacks references to, and comparison with, similar published indices and concepts.**

Authors: The proposed literature by the reviewer is very valuable and will be included in the article, allowing the reader to get a better understanding of the parameters used in the EBFI.

3. **It lacks explanation of the ideas, theory, or reasoning that underlies the EFBI.**

Authors: The concepts required to compute EFBI are mentioned and described in the article. Ideas and reasoning of the proposal are implicit in the formulas and described in paragraph in lines 90 and 155. References to other articles describing or using some of the components included in the EFBI are included in the article, allowing the reader to have a deeper understanding of the underlying factors affecting the behavior of the EFBI, in addition to examples such as that described from line 174 onwards. The overall aim of the paper is to evaluate the quantifiable amount of information that EFBI provides in relation to convection trends in fires on a global scale using reanalysis weather data.

4. **The EFBI is used and promoted as an index for identifying extreme fire behaviour (defined by fire spread rate), but appears to be designed to identify a small subset of those events – fires that produce deep moist convection.**

Authors: EFBI evaluates the convection trend and potential conditional change around the fire using ERA5 reanalysis. It is true that fires driven by convection may be only a subset of all extreme fires, but the EFBI provides the mechanisms to identify these critical events in advance, providing essential information to firefighting services and enhancing the potential response to those fires. Currently, there are neither open and global datasets of fire events classified by type nor exhaustive data from vertical soundings or radar analysis of fire columns that would allow the classification of fire events and all their transitions. Although the FWI can contribute to discriminating fires according to their spread, our results show that EFBI provides an additional contribution to this discrimination on the trend of convection of the atmosphere (also caused by moisture at surface) and conditional instability.

5. **The performance of the EFBI is not convincing.**

Authors: In the original manuscript the NN classification of large events reached 65,46% using EFBI and 58% with FWI only. After the comments of a first reviewer, we modified the NN, reaching a 78,37% with a 1.85% of standard error (included in the answer to M.Pinto not in the manuscript read by the reviewer), and added the C-Haines in the comparison. We acknowledge that EFBI does not explain all extreme events, and this fact is included in our manuscript. However, in our analysis, we show which percentage of fire events (among the selected by the described automatic method) is explained by the EFBI. Most of PyroCb articles suggested by the reviewer focus on single events in which the authors explain a single observed element of the fire. In our analysis, we use the EFBI without any prior knowledge of the fires and it still provides satisfactory results, which corroborates that the ideas exposed by the reviewer are still valid and agree the results hereby presented. The accuracy achieved with the EFBI in classifying fire events at the global scale (78.37%) increases the accuracy obtained with FWI and C-Haines (63.42%), showing that the EFBI provides relevant information in the characterization of fires at that scale. In addition to the analysis at the global scale, in our work we present 3 case studies in Pedrogao (Portugal), Bolivia and Australia, which is more than the cases studied in most of the references suggested by the reviewer.

**The manuscript will require substantial revision to address these concerns. In anticipation of a major rewrite, I decided not to provide a list of minor points to be addressed.**

**Major concerns expanded:**

6. **The paper contains incomplete equations with undefined terms (the EFBI equation is difficult to understand and possibly contains a sign error), and insufficient information in figure captions. A number of terms and concepts are used loosely, which make comprehension difficult for the reader. For example, the term "convective fire" is used almost synonymously with extreme fires. Given all fires produce heat and convection, the term "convective fire" needs to be clearly defined. Also, it is not clear if it refers to fires that produce deep convection columns, or specifically fires that produce moist convection (i.e., pyroCu or pyroCb). Discussions often include references to the very broad topic of "wildfire-atmosphere interactions". For clarity the reader needs to know which specific interactions are being referred to.**

Authors: We thank the reviewer for noting the typo in the EFBI formulas. The typo was in the formula stating the condition CIN <= 0. CIN is negative when there is inhibition, opposite to CAPE. The manuscript was modified stating CIN >= 0 in eq.4, meaning that no convective inhibition exists. The EFBI is based on already well known meteorological concepts such as CAPE and CIN. EFBI estimates the temperature difference at surface that may cause a null or positive CIN. All the required formulae to compute the EFBI are presented in the article. Regarding the term convective fire, it refers to a fire which is driven by convection, as opposed to, for instance, a wind driven fire. As the reviewer mentions, fire produces convection, however fire spread is not always driven by convection as a main factor. We refer to convective a fire when it is driven mostly by convection, not the wind, topography, fuel types, fuel conditions, etc. As suggested by the reviewer, the term "convective fire" has been defined and the definition included in section. We agree with the lack of precision in some of the text in the manuscript. We have further explained the term "wildfire-atmosphere interactions" where it was present in the text.

7. **There are two main topics that need to be referenced, plus a paper that the authors might wish to consider for comparison: (i) In the manuscript the EFBI is claimed to be similar to the Haines indices (Lines 73-75), but a description of these indices (and the often-used modified Haines index, C-Haines) is lacking. Most readers familiar with the Haines indices will not immediately see the similarities. Indeed, the Haines indices were designed to assess atmospheric dryness and absolute stability, whereas the EFBI assesses conditional instability. The Haines indices were developed because existing stability indices used for thunderstorm forecasting were known to be invalid for extreme fires. Given this historical progression, the EFBI design might appear to readers to be**

**regressing. It follows that a comprehensive justification is required to demonstrate that the EFBI is a better discriminator of extreme fire behaviour than the Haines indices, as is implied in the manuscript. (ii) Recent works that develop indices and techniques for identifying and predicting pyroCb (a specific subset of extreme fire events) should also be acknowledged and compared with the EFBI. The EFBI has much in common with Potter's (2005) FireCAPE concept, Lareau and Clement's (2016) use of the Convective Condensation Level, Tory et al.'s (2018) pyroCAPE concept, the ideas that underpin Tory and Kepert's (2021) Pyrocumulonimbus Firepower Threshold (PFT), and Leach and Gibson's (2021) pyrocumulus prediction model. Section 5 of Tory et al. (2018) discusses and compares some of these concepts and ideas. (iii) The authors might like to compare their FWI/EFBI analysis of extreme fire events, with a paper by Di Virgilio et al. (2019) that performed a similar comparison using the McArthur Forest Fire Danger Index and C-Haines to analyse pyroCb events.**

Authors: The authors thank the reviewer for the relevant points mentioned in this comment and the number of references provided in it. As regards the overall message in the comment, we would like to note that the EFBI does not exclusively look into conditioned stability as the EFBI also shows the buoyancy trend. In 1988, Donald Haines provided an extremely useful and pragmatic index to estimate the potential dry and unstable air. Currently, some very severe wildfires had high C-Haines values, although sometimes that only happens when using specific pressure levels different from the ones considered in high, medium, and low versions of C-Haines. In a previous answer to the valuable comment of M. Pinto, we replied by including a figure that depicts that fact in the fire event of Pedrogao (Portugal), where C-Haines index values depict a low sensitivity in all three versions compared with the EBFI. Nowadays we have resources to compute buoyancy trends using all the levels. However, we still lack a database of fire events classified by fire type on that basis, while we only know some fire characteristics such as fire spread. Therefore, we proposed an index that can be efficiently computed on a global scale and exploits the known information about fires. In our results, we state that the EFBI provides valuable information and contributes to enhance the accuracy performance of the classification at global scale despite there is uncertainty in the datasets used due to resolution gaps and cloud coverage affecting GlobFire (based on MCD64A1). Finally, as we described in the paper and also in point number 5 above, there is an enhanced performance of the EFBI when compared to the C-Haines and FWI together or only with the FWI.

The scope of the paper was not detecting pyroCb and therefore the datasets used in the article do not include pyroCb or pyroCu detection. The comparison of the EFBI with other indices, in addition to the C-Haines, which is already considered and included in the new version of the manuscript, is out of the scope of our work. We would like to note that in most of the papers suggested by the reviewer there is not only a lack of comparison with other important works like Haines, but also, the analysis are done at national scale with very few study cases. An exception to this was the work of Di Virgilio et al. (2019), who used two

different indexes and a database which covered the cases within the state of Victoria (Australia) and analyzed 196 cases to discriminate two classes of fires of which 40 were pyroCb fires and 166 were considered as standard wildfires. We found all those papers essential to describe the state of the art of the fire and atmosphere research and cited them in the revised version of the article.

8. **The EFBI is introduced without sufficient explanation. The paper does not describe the underlying theory behind the index, or why the index takes this specific form, or how it compares with similar indices and concepts.**

Authors: The article mixes several research fields and explains the tasks done in each with plain text for a multidisciplinary journal for Natural Hazards and Earth Sciences. In data and methods section we included the following statement: "The proposed EFBI determines the amount of increase in temperature degrees at the surface required to cause a null CIN and quantifies the change in the convective trend (addition of CAPE and CIN), allowing the prediction of fast fire spread due to convection. In cases in which the atmosphere is already unstable, CIN is equal to or greater than 0 being $\Delta T=0$, the values assigned to the index are the full integration of CAPE+CIN." followed by the formulas used to compute it and its explanation. There is also an example in section 3 with a skew-T diagram. The article does not provide any new meteorological concept but uses known concepts applied for fire danger and evaluates the results with two global datasets to see the use feasibility for fire danger under GWIS project. Therefore the work proposes a first version of the index at global scale which is comparable around the planet and ready to use with global datasets. The information provided by EFBI has been used in fire events in Europe and California during 2020 and 2021.

9. **The type of convective instability being targeted is not mentioned. The EFBI looks like it is targeting moist convection, but it is used to identify any fire that spreads rapidly. Can the decision to apply a moist-instability index to dry events be justified? In general, hot, dry and windy conditions favour extreme fire behaviour. Large values of the Haines indices correlate well with deep well-mixed layers, which have neutral stability on average. Moist plume growth, on the other hand, requires a much more specific set of conditions and consideration of the atmosphere above the mixed-layer.**

Authors: The article mentions already that extreme fire behavior is not just caused by convection (line 276). The aim of the work is not modelling the fire plume, but a first approach including convection trend evaluating the usefulness of the ERA5 and GlobFire at global scale. EFBI assesses convection trend using the vertical integral of the buoyancy and how it would change with an artificial increase of the temperature at surface (defined in section 2). We agree with the reviewer that the work does not justify the decision to apply EFBI for fires not driven by convection. However, results show a considerable increase in accuracy in the classification that when using CHaines+FWI or only FWI. It would be a major improvement to use a global database with a good accuracy of classification between spread types for each

moment of the fire. The study of detailed moist plume growth, transition between wind driven fires and convective and viceversa would require field measurements like doppler radar or lidar.

10. **The manuscript doesn't present compelling evidence that the EFBI has significant value as an extreme-fire predictor. The scatter plot in Fig. 8 shows that large EFBI occurs more often with slow spread rates than fast spread rates. 7 shows EFBI is elevated throughout almost the entire fire period, but the extreme fire behaviour was present only on the first day. Fig. 10 shows poor correlation between EFBI and the burn rates – which is acknowledged by the authors. Fig. 12 confirms southeastern Australia was very unstable during the last few days of 2019 – a result that was also well predicted by the C-Haines and PFT indices.**

Authors: The EFBI shown in the results a considerable amount of improvement to discern between fires that spread less than 1000ha and more than 10000ha in one day. It has been compared with FWI and with FWI combined with C-Haines using a machine learning approach using two main datasets and a cross-validation. The accuracy performance of the EFBI and the variability in the cross validation is included in the manuscript and in the answer provided to M. Pinto. In addition, there are more factors involved in extreme fire events, fuel availability, vertical wind profile field measurements, etc. Figure 7 shows a considerable peak of the EFBI. We conclude that EFBI could be a necessary but not sufficient condition as happens with many other fire danger indices. Figure 7 shows fire spread speed below 1 km/h only when EFBI is below 200. Figure 10 is using GlobFire for the fire spread speed (one perimeter per day with uncertainty) and is using forecast data. The fire in Robore burnt for more than two months and was visible when looking at the entire Earth. Robore fire case transitioned from wind driven to convection driven and vice versa several times. The last case study shows how the daily maximum of the EFBI behaves spatially during the fire events in south-eastern Australia. As the reviewer mentioned, EBFI alone is not enough to detect sufficient premises for a pyroCb. However, EFBI, vertical wind profile and the rest of the fire context can be used to assess periods of extreme fire behavior. For real firefighting planning, EFBI has been used in Dixie fire with a time horizon of 10 days ahead meanwhile Tory's PFT is 24 hours ahead. Both useful, but different goals and application. EFBI leads to a planning measure for extreme fire spread at long term always keeping in mind the limitations and uncertainties.

**In summary, the manuscript is rough in presentation, is missing important background information, introduces a concept without sufficient explanation and justification, and the results as presented do not convince me that the EFBI has more to offer than existing indices and extreme-fire prediction methods. I encourage the authors to consider the ideas presented in the referenced papers to see if the EFBI can be adapted to better identify extreme fire conditions, or to develop arguments that demonstrate EFBI superiority over these other indices.**

Authors: The aim of the article is not to show superiority or validate other works but evaluate the usefulness of the proposal on a global scale using open data. The data used in this work has limitations, uncertainties and there is a lot of future work as stated in the paper. We could apply only FWI to detect extreme fire behavior, but we decided to quantify and evaluate the performance of a feasible method to assess convective trend at global scale. The work is a first and pragmatic approach to use a global fire event database and ERA5 weather reanalysis trying to be distinguish only two fire types by daily speed. Therefore, the article is more focused on analyzing an approach at global scale with the available data instead of creating, analyzing causes of pyroCb or comparing with previous proposals in the literature. Finally, several of the key and exceptionally good works provided by the reviewer do not evaluate performance on a global scale, validate Haines or show superiority over all previous indexes. Each work is focused on different goals and provides extremely valuable contributions. Given the distribution of fire events by speed, where slow fires are still the most common ones, it would be extremely useful to provide useful open data to countries which are starting to have these extreme fire events that are still considered as a rare phenomenon.

**Reviewer 2:**

**References**

- Di Virgilio G, Evans J.P, Blake S.A.P, Armstrong M, Dowdy A.J, Sharples J and McRae R. 2019. Climate Change Increases the Potential for Extreme Wildfires. Geophysical research Letters, 46. <https://doi.org/10.1029/2019GL083699>
- Lareau, N. P., and C. B. Clements, 2016: Environmental controls on pyrocumulus and pyrocumulonimbus initiation and development. Atmos. Chem. Phys., 16, 4005–4022, https://doi.org/10.5194/acp-16-4005-2016.
- Leach, R. N, and C. V. Gibson, 2021: Assessing the potential for pyroconvection and wildfire blow ups. J. Operational Meteor., 9 (4), 47-61, doi: https://doi.org/10.15191/nwajom.2021.0904.
- Potter, B. E., 2005: The role of released moisture in the atmospheric dynamics associated with wildland fires. Int. J. Wildland Fire, 14, 77–84, https://doi.org/10.1071/WF04045.
- Tory, K. J., W. Thurston, and J. D. Kepert, 2018: Thermodynamics of pyrocumulus: A conceptual study. Mon. Wea. Rev., 146, 2579–2598, https://doi.org/10.1175/MWR-D-17-0377.1.
- Tory, K. J., and J. D. Kepert, 2021: Pyrocumulonimbus Firepower Threshold: Assessing the atmospheric potential for pyroCb. Weather and Forecasting., 36, 441—456, https://doi.org/10.1175/WAF-D-20-0027.1.

**Citation**: https://doi.org/10.5194/nhess-2021-122-RC2

---

## Author Response (AR2)

**2n Round**

Editor:

08 Nov 2021

**Editor decision: Reconsider after major revisions (further review by editor and referees)**

by Ricardo Trigo

**Comments to the author**:
Dear Authors,

Two reviewers have now provided a detailed review of the revised manuscript. Both appreciate your effort and overall improvements of the revised version (when compared to the submitted manuscript). Nevertheless, both reviewers still require further revisions. In particular reviewer#2 has a substantial amount of criticisms and suggestions. Therefore I would like to invite you to submit a revised version of your manuscript.

Please provide an 'author's reply' to each issue raise by both reviewers. Please can you also include a track changes document between the old manuscript and the new one (you can include this as part of your 'author's reply').

Looking forward to see a much improved version

Best regards

Ricardo Trigo

Authors:

Dear reviewers and editor,

We express our gratitude for the work and time dedicated to this review that improved the article. We answered and modified accordingly the points raised by the reviewers.

Best regards,

**Reviewer Reports:**

| Report #1 | |
|---|---|

Submitted on 28 Oct 2021
Referee #1: Miguel Pinto, miguelotnip@hotmail.com

| Anonymous during peer-review: | Yes | **No** | |
|---|---|---|---|
| Anonymous in acknowledgements of published article: | Yes | **No** | |

**Recommendation to the editor**

| | | | | |
|---|---|---|---|---|
| **1) Scientific significance** Does the manuscript represent a substantial contribution to the understanding of natural hazards and their consequences (new concepts, ideas, methods, or data)? | **Excellent** | Good | Fair | Poor |
| **2) Scientific quality** Are the scientific and/or technical approaches and the applied methods valid? Are the results discussed in an appropriate and balanced way (clarity of concepts and discussion, consideration of related work, including appropriate references)? | Excellent | Good | **Fair** | Poor |
| **3) Presentation quality** Are the scientific data, results and conclusions presented in a clear, concise, and well-structured way (number and quality of figures/tables, appropriate use of technical and English language, simplicity of the language)? | Excellent | **Good** | Fair | Poor |

For final publication, the manuscript should be

| |
|---|
| **accepted as is**. |
| accepted subject to **technical corrections**. |
| accepted subject to **minor revisions**. |
| **reconsidered after major revisions:** |
| rejected. |

| |
|---|
| **Were a revised manuscript to be sent for another round of reviews:** |
| **I would be willing to review the revised manuscript.** |
| I would not be willing to review the revised manuscript. |

| Suggestions for revision or reasons for rejection (will be published if the paper is accepted for final publication) |
| --- |

The authors answered all my comments regarding the first version of the manuscript and improved the manuscript accordingly. In particular, the authors included the Continuous Haines index in the analysis and the results of the MLP training suggest a significant advantage for the EFBI over the Continuous Haines. There are however important aspects that are missing or need clarification.

In particular, the Continuous Haines is not mentioned in the Data and Methods section. In fact, it is only mentioned in L222 when describing the accuracy of the MLP and it was apparently added to Figure 4 without any commentary.

Added to methodology (levels do we use for the two components of the CHaines).

The results of the MLP are very good with an accuracy of 78% when using the EFBI combined with the FWI components percentiles ("FWI" + EFBI) but they need a better justification to be convincing. Particularly so if a change in the model architecture and optimizer improved this number from 65% (in the first version of the manuscript) to 78%. This suggests that the MLP in the first version didn't converge to an optimal solution. I didn't find in the text if the data is normalized/standardized before applying the MLP. If not, I suggest that the authors should repeat the MLP training for the 3 experiments ("FWI", "FWI" + EFBI, "FWI" + CHI) with data normalized to have a mean of 0 and standard deviation of 1. For the decision tree, I suggest presenting the results for the ("FWI" and "FWI" + CHI) cases since only the "FWI" + EFBI case is presented. The comparison would be relevant for the discussion.

The optimizer cannot guarantee optimal solution even using a hyperparameter tunning for each experiment. The work is a first approach that shows the importance of an atmosphere-wildfire index. The 3 experiments are already as the reviewer suggest "FWI" 60.75% acc., "FWI" + EFBI 78.37% acc, "FWI" + CHI 63.42% using normalization for all of them, (also added to the manuscript). The decision tree is used to illustrate idea about the value of the results in the mutual information when combined with other important input parameters as FWI.

Finally, a comparison of the MLP results for the 3 experiments ("FWI", "FWI" + EFBI, "FWI" + CHI) in a visual format would bring further value to the discussion of the results. One suggestion would be to use the outputs of the MLP in probability form and make histogram plots for the distribution of the probabilities in the two classes (small, large fires) and for the 3 experiments.

Agree with the reviewer, EFBI would represent one component to a future fire danger. Assuming that the proposal can be further developed, we must follow working as the reviewer mentions using a probabilistic approach from the last layer of the NN but also increase the number of classes. It is also true that danger classes cannot be just small and large, we expect to be able to reduce the variability of the accuracy, retrieve better information and increase the number of classes to reach a useful final fire index which included severe fire behavior. Therefore, a more developed proposal would use a probabilistic approach and a confusion matrix for a given fire in addition to the final classification. The proposed work shows the relevance of atmosphere data

for severe fire danger, but this proposal must be improved using fire types and new classes for the classification to be used only with a ML approach. The ML approach is used to point out that EFBI can provide information but the available datasets for the assessment include limitations that make it hard to go further in a single article. With a refined dataset and less uncertainty more complex ML approaches can be applied using hyperparameter tunning.

Additional comments:

L156 (previous L136) - I think the wording on this sentence is what can make it confusing. The sentence says "the initial day of the time window (...) was increased by two days" which I interpret as: the initial day was moved forward by two days hence the time window reduced. In the answer to my previous comment, the authors wrote "We increased the time window" which I interpret as the initial day being decreased by two days to account for weather conditions of fires with dates mapped later than the correct day. I suggest rewording the sentence to avoid confusion.

Done.

L213-214 - I would expect to see some comments regarding the MI comparison for the EFBI and the Continuous Haines. At which level is the Continuous Haines computed? Or is the level selected based on the height at the location of the event? A MI of 0 for CH_max may suggest saturation of this index, can the authors provide some comment on that?

Done

L219-220 - This sentence may also need to be restructured. For instance, the ReLU activation function is part of the model architecture, it is not related to the Adam optimizer as the text may suggest.

Changed

Overall the results are promising but the aspects mentioned above need to be addressed to better convey the results and to make sure they can be reproduced.

Authors really appreciate the suggestions of the reviewer, which are extremely useful and improve the work. We consider that the probabilistic approach is the next following step for an improvement of the proof of concept and potential usefulness. But for that, we need a full human validated dataset of fire classes as mentioned in the conclusions. Levels for the CHaines have been included and now is also specified in the text the normalization of the inputs for the ML for the three experiments.

Submitted on 04 Nov 2021
Anonymous Referee #2

| Anonymous during peer-review: | Yes | No |
|---|---|---|
| Anonymous in acknowledgements of published article: | Yes | No |

**Recommendation to the editor**

| | Excellent | Good | Fair | Poor |
|---|---|---|---|---|
| **1) Scientific significance** **Does the manuscript represent a substantial contribution to the understanding of natural hazards and their consequences (new concepts, ideas, methods, or data)?** | Excellent | Good | **Fair** | Poor |
| **2) Scientific quality** **Are the scientific and/or technical approaches and the applied methods valid? Are the results discussed in an appropriate and balanced way (clarity of concepts and discussion, consideration of related work, including appropriate references)?** | Excellent | Good | Fair | **Poor** |
| **3) Presentation quality** **Are the scientific data, results and conclusions presented in a clear, concise, and well-structured way (number and quality of figures/tables, appropriate use of technical and English language, simplicity of the language)?** | Excellent | Good | **Fair** | Poor |

For final publication, the manuscript should be

accepted as is.

accepted subject to **technical corrections**.

accepted subject to **minor revisions**.

**reconsidered after major revisions:**

rejected.

**Were a revised manuscript to be sent for another round of reviews:**

**I would be willing to review the revised manuscript.**

I would not be willing to review the revised manuscript.

**Suggestions for revision or reasons for rejection (will be published if the paper is accepted for final publication)**

The authors have made some changes to the manuscript, but my major concerns remain. I suspect I did not express these concerns clearly enough. I will begin this review by paraphrasing and commenting on the five central points of my original review:

1. Presentation: The presentation has improved. There still might be a sign error in the EFBI equation (it looks like it will yield negative values). Confusion remains regarding wind-driven and

plume-driven fires etc. (see below). I still feel there needs to be more detail included in some of the figure captions. For example, the Convective Condensation Level (CCL), which appears to be key to the EFBI, is marked in Fig. 2, but there is no mention of it in the text or caption.

We really appreciate the work done by the reviewer. The difference in the manuscript in the main formula (eq1.) was wrong, the other way around. It should be the altered state (CAPEP+CINP) less the original state (CAPE+CIN) according how the computation is done. We apologize for the confusion generated with two consecutive mistakes in the formulas.

About the negative value of the EFBI:
CAPEP+CINP-CAPE-CIN → CAPE is greater than 0 by definition but also less than CAPEP which is also greater than 0. CINP is 0 or a small positive. While CIN is negative or 0 by definition.  So, CAPEP-CAPE >= 0 and CINP-CIN >= 0.

2. Lack of references to similar published indices: The suggested papers are now mentioned, but there is no discussion of the indices and how they compare with the EFBI. A more detailed explanation of the EFBI and its illustration in Fig. 2, would provide opportunities to reference some of the concepts introduced in these papers, and clear up confusion elsewhere in the text (see the 'Additional points of confusion' below). For example, the CCL appears to be a concept central to the EFBI (the CAPEP parcel emerges from the CCL). This would be a good opportunity to explain the EFBI in more detail, and to acknowledge Lareau and Clements work, which uses the CCL is as a pyroCb predictor. Another example is the close similarity between the EFBI ΔT term, and Leach and Gibson's ΔT term and Tory and Kepert's Δθ term.

Absolutely agree with the reviewer, a paragraph including the assumptions and the physical approach has been added referencing the similarities with previous works. However, we cannot consider that the EFBI is a fully realistic physical approach. We really appreciate the work, time, and patience of the reviewer.

3. The EFBI is presented with insufficient physical justification: The EFBI is described, but no reasoning is given for why this particular function was chosen. My concern here remains. See also the previous point.

This work is not proposing a model, but an indicator based on data that does not see the fire and is evaluated at global scale. We clarified that the index focuses on moist convection and the similarity with ΔT term. Also, we changed the text, and before the description of the EFBI and the equations:

"The extreme fire behaviour index(EFBI) determines the amount of increase in temperature degrees at the surface required to cause a null CIN and quantifies the change in the available convective energy. The amount of change of convective energy per degree is used as indicator for deep moist convection."

4. The EFBI is promoted as a tool for identifying extreme fire behaviour but is designed to identify moist convection: The authors msunderstood my concern in this point (see below).

The text has changed to specify moist convection.

5: The EFBI performance is not convincing: This is my opinion based on seeing other tools in operational use. Performance assessments of those other tools have not been published. Hence, I concede that my criticism on this point was unfair.

Main point of confusion.
Historically, fires have been classified as wind-driven or plume-driven, with anecdotal evidence suggesting plume-driven fires are associated with rapid fire spread and unpredictable fire behaviour. Wind-driven fires are loosely classified as fires with flames blown by the wind, igniting fuel by direct contact. Plume-driven fires have upright convection columns, with inflow from all directions, with radiation heating the fuel to ignition. This simple distinction continues to be debated.
A relationship between upright convection columns and dry convective instability was established decades ago. Dry convective instability occurs typically in deep, neutral mixed layers, and the instability is caused by surface radiative heating.

Haines developed a very simple measure for diagnosing deep, neutral and dry mixed layers. Being very simple, it has limitations. The Haines indices fail when the pressure levels, hard-wired into the indices, do not adequately sample the mixed-layer. Another limitation is that they saturate for moderately deep and dry mixed layers.

Very deep convection columns in the right atmospheric conditions can lead to cloud formation (pyrocumulus, pyroCu) and thunderstorms (pyrocumulonimbus, pyroCb). These can introduce additional hazards. It follows that conditions that support PyroCb development are a subset of conditions that support convection-column development.
Fire CAPE, Convective Condensation Level, Pyrocumulonimbus Firepower Threshold, and Leach and Gibson's tools, are all designed to identify this specific subset of conditions, i.e., pyroCu/pyroCb.

The EFBI is a modified CAPE index, and CAPE is a thunderstorm diagnostic. Large values of EFBI identify conditions where a relatively small amount of heating generates a large increase in conditional instability, i.e., a large increase in thunderstorm potential. By design it does not target dry convection.

The authors imply that the EFBI is an index designed to identify conditions that support convection-column development in general, which is incorrect.

I suggest the text be adjusted so that it acknowledges the index was designed to target moist instability (thunderstorm potential). The fact that it seems to work when no cloud is observed in the plume, is largely accidental. I suspect this is because small ΔT will be present on typical days of high fire danger.

Text has been changed specifying EBFI targets deep moist convection. EFBI does not assess dry convection by design, any effect of dry convection in the EFBI time evolution depends on the input data used, in this case from IFS model used in ERA5.

Additional points of confusion.
Some of the descriptions surrounding how the EFBI works are misleading. There appears to be a misunderstanding of how fires interact with the atmosphere. The second paragraph in section 3

introduces the concept of CAPE, and notes that a hypothetical surface-temperature increase (here 14 C) can reduce CIN and increase CAPE, thereby causing "a considerable change in the vertical air motion" and "a stable atmosphere can become unstable". The fire adds 300-500 C of surface heating, which is more than 20 times the 14 C mentioned in the example. The combustion gases are very hot, and thus the smoke plume near the surface is extremely unstable. All wildfire smoke plumes are extremely unstable. These fires do not change the stability of the atmosphere surrounding the plumes by any measurable amount. Thus, the sentence is irrelevant in the wildfire context.

Sentence changed. Added clarification of the 14 C increase. We do not estimate fire heat release, parcel simulation or any use of high frequency fire monitoring, biomass estimation or gas emission. We cannot estimate those values accurately for all fire events and how they may affect the stability of the atmosphere on a global scale. Only a hypothetical change in temperature as an indicator, which is not directly the energy released by the fire, may cause a change in the atmosphere.

As the plume rises it is rapidly diluted when cooler air from outside is mixed in to the plume. If the plume cooling is not too rapid, the plume may rise high enough to condense (where, in the quoted example, it will be 14 C or more warmer than the surface air), and then the added buoyancy from condensational heating (diagnosed by CAPEP) can be realized. But there are many factors that influence plume entrainment (e.g., the fire shape, size and intensity, the wind speed, the absolute atmospheric stability, to name a few) most of which are independent of the surface temperature. In fact, recent arguments suggest the surface temperature (and humidity) is largely irrelevant because most of the air in the plume has been entrained from layers above the surface (e.g., Tory and Kepert 2021). If this is true, the next paragraph (line 188-191) is also problematic.

CAPE, CIN are used as stability indicators too and are not independent of the surface temperature and dew point temperature for a given timestep. Both are used to find LCL, using the moist adiabatic lapse rate and temperature profile is used to find LFC and EL used for CAPE and CIN. So, surface temperature and dew point also affect CAPE and CIN. We agree with the reviewer that CAPE and CIN values are susceptible to being inaccurate because of the estimation of values at surface and that the air of the plume can be entrained from upper layers. But also, there are fire cases that start with low height plumes followed by sudden blow up with an increase of plume altitude. But the datasets we have available cannot be used to validate or evaluate the entrainment ease of air of the plume into upper layers. It is clearly one of the next factors to tackle learning from models as the one described by Tory and Kepert.

The sentence (line 185-186) "A wildfire in this condition has a high likelihood of becoming driven by convection" is also misleading. Presumably, the authors are referring to the plume-driven fire concept (as distinct from wind-driven) associated with a deep convection column. But before the moist instability can be released the smoke plume must rise typically 3—5 km, in which case it must have been convectively driven for some considerable time (to get to that height). Clearly, the potential for moist instability is irrelevant for establishing an environment that promotes plume-driven fires, which gets back to the main point of confusion.

In short, the EFBI is moist-convection diagnostic that only has relevance to a smoke plume that manages to rise 3—5 km, but the text references surface-based instability to describe the

instability.

I recommend the authors consider review articles on fire dynamics, before editing the manuscript. Good examples include:
Sullivan, A., 2017: Inside the Inferno: Fundamental Processes of Wildland Fire Behaviour Part 1: Combustion Chemistry and Heat Release. Current Forestry Reports.
https://link.springer.com/article/10.1007/s40725-017-0057-0
Sullivan, A., 2017: Inside the Inferno: Fundamental Processes of Wildland Fire Behaviour Part 2: Heat Transfer and Interactions. Current Forestry Reports.
https://link.springer.com/article/10.1007/s40725-017-0058-z
Dominique Morvan, Nicolas Frangieh. Wildland fires behaviour: wind effect versus Byram's convective number and consequences upon the regime of propagation. International Journal of Wildland Fire, CSIRO Publishing, 2018, 27 (9), pp.636. ff10.1071/Wf18014ff. ffhal-02114689f

Additional responses are included below, prefaced by "Reviewer:".

2. It lacks references to, and comparison with, similar published indices and concepts.
The proposed literature by the reviewer is very valuable and will be included in the article allowing the reader the better understanding of the parameters used in the EBFI.

Reviewer: The article lists the additional papers. It does not discuss the tools/indices proposed in those papers in any detail, or compare them with the EFBI. The existing tools could provide context for the EFBI.

3. It lacks explanation of the ideas, theory, or reasoning that underlies the EFBI.
The concepts required to compute EFBI are mentioned and described in the article. Ideas and reasoning of the proposal are implicit in the formulas and described in paragraph in line 90 and 155. References to other articles describing or using some of the components included in the EFBI are included in the article, allowing the reader to have a deeper understanding of the underlying factors affecting the behavior of the EFBI in addition to examples like the description that starts in line 174. The overall aim of the paper is to evaluate the quantifiable amount of information that EFBI provides in relation to convection trends in fires on a global scale using reanalysis weather data.

Reviewer: See the "Additional points of confusion" described above.

4. The EFBI is used and promoted as an index for identifying extreme fire behaviour (defined by fire spread rate), but appears to be designed to identify a small subset of those events – fires that produce deep moist convection.
EFBI evaluates the convection trend and potential conditional change around the fire using ERA5 reanalysis. It is true that fires driven by convection may be only a subset of all extreme fires, but the EFBI provides the mechanisms to identify these critical events in advance, providing essential information to firefighting services and enhancing the potential response to those fires. Currently,

there are neither open and global datasets of fire events classified by type nor exhaustive data from vertical soundings or radar analysis of fire columns that would allow the classification of fire events and all their transitions. Although the FWI can contribute to discriminating fires according to their spread, our results show that EFBI provides an additional contribution to this discrimination on the trend of convection of the atmosphere (also caused by moisture at surface) and conditional instability.

Reviewer: See the two discussions above on points of confusion.

The manuscript will require substantial revision to address these concerns. In anticipation of a major rewrite, I decided not to provide a list of minor points to be addressed.

Major concerns expanded:
6. The paper contains incomplete equations with undefined terms (the EFBI equation is difficult to understand and possibly contains a sign error), and insufficient information in figure captions. A number of terms and concepts are used loosely, which make comprehension difficult for the reader. For example, the term "convective fire" is used almost synonymously with extreme fires. Given all fires produce heat and convection, the term "convective fire" needs to be clearly defined. Also, it is not clear if it refers to fires that produce deep convection columns, or specifically fires that produce moist convection (i.e., pyroCu or pyroCb). Discussions often include references to the very broad topic of "wildfire-atmosphere interactions". For clarity the reader needs to know which specific interactions are being referred to.

We thank the reviewer for noting the typo in the EFBI formulas. The typo was in the formula stating the condition CIN <= 0. CIN is negative when there is inhibition, opposite to CAPE. The manuscript was modified stating CIN >= 0 in eq.4, meaning that no convective inhibition exists. The EFBI is based on already well known meteorological concepts such as CAPE and CIN. EFBI estimates the temperature difference at surface that may cause a null or positive CIN. All the required formulae to compute the EFBI are presented in the article. Regarding the term convective fire, it refers to a fire which is driven by convection, as opposed to, for instance, a wind driven fire. As the reviewer mentions, fire produces convection, however fire spread is not always driven by convection as a main factor. We refer to convective a fire when it is driven mostly by convection, not the wind, topography, fuel types, fuel conditions, etc. As suggested by the reviewer, the term "convective fire" has been defined and the definition included in section. We agree with the lack of precision in some of the text in the manuscript. We have further explained the term "wildfire-atmosphere interactions" where it was present in the text.

Reviewer: The definitions used to explain the fire type remain confusing. Perhaps use the term "convection column" when describing the plume type being targeted. See discussions above.

7. There are two main topics that need to be referenced, plus a paper that the authors might wish to consider for comparison: (i) In the manuscript the EFBI is claimed to be similar to the Haines indices (Lines 73-75), but a description of these indices (and the often-used modified Haines index, C-Haines) is lacking. Most readers familiar with the Haines indices will not immediately see the similarities. Indeed, the Haines indices were designed to assess atmospheric dryness and absolute stability, whereas the EFBI assesses conditional instability. The Haines indices were developed because existing stability indices used for thunderstorm forecasting were known to be invalid for extreme fires. Given this historical progression, the EFBI design might appear to readers to be regressing. It follows that a comprehensive justification is required to demonstrate that the EFBI is a better discriminator of extreme fire behaviour than the Haines indices, as is implied in the

manuscript. (ii) Recent works that develop indices and techniques for identifying and predicting pyroCb (a specific subset of extreme fire events) should also be acknowledged and compared with the EFBI. The EFBI has much in common with Potter's (2005) FireCAPE concept, Lareau and Clement's (2016) use of the Convective Condensation Level, Tory et al.'s (2018) pyroCAPE concept, the ideas that underpin Tory and Kepert's (2021) Pyrocumulonimbus Firepower Threshold (PFT), and Leach and Gibson's (2021) pyrocumulus prediction model. Section 5 of Tory et al. (2018) discusses and compares some of these concepts and ideas. (iii) The authors might like to compare their FWI/EFBI analysis of extreme fire events, with a paper by Di Virgilio et al. (2019) that performed a similar comparison using the McArthur Forest Fire Danger Index and C-Haines to analyse pyroCb events.

The authors thank the reviewer for the relevant points mentioned in this comment and the number of references provided in it. As regards the overall message in the comment, we would like to note that the EFBI does not exclusively looks into conditioned stability as the EFBI also shows the buoyancy trend.

Reviewer: The EFBI targets conditional instability by design (it is a function of CAPE, a conditional instability diagnostic). The fact that it works for dry convection is largely accidental. See discussions above.

In 1988, Donald Haines provided an extremely useful and pragmatic index to estimate the potential dry and unstable air. Currently, some very severe wildfires had high C-Haines values, although sometimes that only happens when using specific pressure levels different from the ones considered in high, medium, and low versions of C-Haines. In a previous answer to the valuable comment of M. Pinto, we replied by including a figure that depicts that fact in the fire event of Pedrogao (Portugal), where C-Haines index values depict a low sensitivity in all three versions compared with the EBFI. Nowadays we have resources to compute buoyancy trends using all the levels.

Reviewer: Yes, now that we are able to use information at all levels, we can develop more sophisticated diagnostics. This includes improved diagnostics for identifying dry instability, which would also perform better than C-Haines.

However, we still lack a database of fire events classified by fire type on that basis, while we only know some fire characteristics such as fire spread. Therefore, we proposed an index that can be efficiently computed on a global scale and exploits the known information about fires.

Reviewer: An improved dry-instability index would better exploit our knowledge of fire behavior. The EFBI is irrelevant for many cases of extreme fire behavior. See discussions above.

In our results, we state that the EFBI provides valuable information and contributes to enhance the accuracy performance of the classification at global scale despite there is uncertainty in the datasets used due to resolution gaps and cloud coverage affecting GlobFire (based on MCD64A1). Finally, as we described in the paper and also in point number 5 above, there is an enhanced performance of the EFBI when compared to the C-Haines and FWI together or only with the FWI.

The scope of the paper was not detecting pyroCb and therefore the datasets used in the article do not include pyroCb or pyroCu detection.

Reviewer: While the intention may not have been to detect pyroCu/PyroCb, the EFBI has a strong pyroCb focus. See discussions above.

The comparison of the EFBI with other indices, in addition to the C-Haines, which is already considered and included in the new version of the manuscript, is out of the scope of our work.

Reviewer: My suggestion seems to have been misunderstood. I agree there is no need to test the other indices. I was only suggesting their design and underlying philosophies be discussed, noting similarities and differences.

We would like to note that in most of the papers suggested by the reviewer there is not only a lack of comparison with other important works like Haines, but also, the analysis are done at national scale with very few study cases. An exception to this was the work of Di Virgilio et al. (2019), who used two different indexes and a database which covered the cases within the state of Victoria (Australia) and analyzed 196 cases to discriminate two classes of fires of which 40 were pyroCb fires and 166 were considered as standard wildfires. We found all those papers essential to describe the state of the art of the fire and atmosphere research and cited them in the revised version of the article.

8. The EFBI is introduced without sufficient explanation. The paper does not describe the underlying theory behind the index, or why the index takes this specific form, or how it compares with similar indices and concepts.
The article mixes several research fields and explains the tasks done in each with plain text for a multidisciplinary journal for Natural Hazards and Earth Sciences. In data and methods section we included the following statement: "The proposed EFBI determines the amount of increase in temperature degrees at the surface required to cause a null CIN and quantifies the change in the convective trend (addition of CAPE and CIN), allowing the prediction of fast fire spread due to convection.

Reviewer: This sentence demonstrates poor understanding of fire behavior, convection, and the relationship between fire-spread and moist convection in smoke plumes. See the discussions above.

In cases in which the atmosphere is already unstable, CIN is equal to or greater than 0 being $\Delta T=0$, the values assigned to the index are the full integration of CAPE+CIN." followed by the formulas used to compute it and its explanation. There is also an example in section 3 with a skew-T diagram. The article does not provide any new meteorological concept but uses known concepts applied for fire danger and evaluates the results with two global datasets to see the use feasibility for fire danger under GWIS project.

Reviewer: The article adapts a meteorological concept into a fire-danger tool, the EFBI. I believe the physical justification is largely irrelevant to its stated purpose. See the discussions above.

Therefore the work proposes a first version of the index at global scale which is comparable around the planet and ready to use with global datasets. The information provided by EFBI has been used in fire events in Europe and California during 2020 and 2021.

Reviewer: The index may prove to have some predictive value, which is why I do not wish to recommend rejection. However, the paper shouldn't be published until the inaccurate statements

about what the index is, and misleading statements about plume-atmosphere interactions, have been addressed.

9. The type of convective instability being targeted is not mentioned. The EFBI looks like it is targeting moist convection, but it is used to identify any fire that spreads rapidly. Can the decision to apply a moist-instability index to dry events be justified? In general, hot, dry and windy conditions favour extreme fire behaviour. Large values of the Haines indices correlate well with deep well-mixed layers, which have neutral stability on average. Moist plume growth, on the other hand, requires a much more specific set of conditions and consideration of the atmosphere above the mixed-layer.
The article mentions already that extreme fire behavior is not just caused by convection (line 276). The aim of the work is not modelling the fire plume, but a first approach including convection trend evaluating the usefulness of the ERA5 and GlobFire at global scale. EFBI assesses convection trend using the vertical integral of the buoyancy and how it would change with an artificial increase of the temperature at surface (defined in section 2).

Reviewer: As discussed earlier, this is largely irrelevant.

We agree with the reviewer that the work does not justify the decision to apply EFBI for fires not driven by convection. However, results show a considerable increase in accuracy in the classification that when using CHaines+FWI or only FWI. It would be a major improvement to use a global database with a good accuracy of classification between spread types for each moment of the fire.

Reviewer: I hope the authors can fix the inaccuracies so that the work can be published.

Specific points:
Line 48: "In this work we will refer to fire driven by convection when convection is an important driving factor of the fire spread." This is too vague. The "convection" definition is a critical concept in this paper. What sort of convection is it referring to? How does this convection drive fire-spread? Convection describes the motion of fluid driven by density gradients (i.e., buoyancy). In the wildfire context there are three types of convection commonly discussed: (i) dry convection describing the deep overturning circulations caused by solar heating of the surface in a neutral boundary layer (the focus of the Haines indices), (ii) moist convection, associated with thunderstorms in the smoke plumes (the focus of FireCAPE, CCL, PFT, Leach and Gibson tools), and (iii) convection in the flame-zone and smoke plume. Fire-spread is accelerated mostly by strong winds and dry fuels. The link between any of these forms of convection and fire-spread is not immediately obvious, and arguments I am aware of are mostly anecdotal. The assumptions and hypotheses that underpin the EFBI need to be clearly communicated here.

Lines 50—54: The discussion of the Haines index here would benefit by describing which "atmospheric instability" it was targeting (absolute rather than conditional), and how it contributes to dangerous fire spread. It should note also that Haines was targeting dry instability. His indices have high values during conditions of weak, absolute stability. He had to eliminate humid conditions with weak absolute stability because they are often associated with heavy rain. This is why the Haines indices contain a dewpoint depression term.

Line 63—65: See the discussion above on how plumes trigger thunderstorms, and my argument that surface conditions are largely irrelevant.

Line 76—79: These sentences are bold claims that are not demonstrated in the paper.

Line 103: There still seems to be a sign error in this equation. CINP by definition will be close to zero, and CAPEP will always be larger than CAPE+CIN, which would mean this EFBI equation is negative.

Line 123—124: "Under these conditions, air can potentially move vertically creating local conditions which are not explicitly provided by meteorological forecasts." Meteorological forecasts are very good at identifying conditions that produce vertical motion. High-resolution forecast models are capable of resolving a range of convective circulations. Is the sentence referring to the smoke plume in some way?

Line 125—127: This sentence appears to be out of place. How does it tie in to the previous sentences in the paragraph?

Line 181—184: The discussion around Fig. 2 needs clarification. Add more labels to the figure, explain the labels in the caption, and talk the reader through the process in more detail.

Line 193—194: It might be worth explaining why the EFBI only needs to be large at the beginning of the period when "the fire had extreme behaviour".

Line 270—272: It would surprise me if ERA5 knew anything about the fire. The sentence appears to imply that the fire is affecting the environment in some way that changes how the EFBI functions. The EFBI is designed to diagnose conditions unaffected by the fire, so it should not be a problem if ERA5 knows nothing about the fire. Perhaps I have misunderstood the intended meaning.

Line 289—290: This sentence raises the concern that ERA5 knows about the fire, whereas forecast models will not know about the fire. As in the previous point I would be surprised if ERA5 knows anything about the fire. There would need to be a number of observations near the fire included in the assimilation. Even then, if the observations differ too much from expected values they get rejected after failing quality control. Smoke can affect the local meteorology, but it is not assimilated into any numerical weather prediction system that I am aware of.

Line 294—295: The wording of this sentence could be improved. It suggests the fire interacts with an abstract concept: "atmospheric vertical profile".

Line 299—300: The meaning of this sentence is unclear. Is it suggesting that model surface data could be improved

---

## Author Response (AR3)

**Answer to reviewers NHESS article 122 17 January 2022**

**Dear Editor and Reviewers,**

**We would like to express our gratitude for the work done from the editor and reviewers for this publication which clearly improved the article.**

**Best regards,**
**Tomas Artes**

Referee #1: Miguel Pinto, miguelotnip@hotmail.com

| | | |
|---|---|---|
| **Anonymous during peer-review:** | Yes | **No** |
| **Anonymous in acknowledgements of published article:** | Yes | **No** |

**Recommendation to the editor**

| | | | | |
|---|---|---|---|---|
| **1) Scientific significance** Does the manuscript represent a substantial contribution to the understanding of natural hazards and their consequences (new concepts, ideas, methods, or data)? | **Excellent** | Good | Fair | Poor |
| **2) Scientific quality** Are the scientific and/or technical approaches and the applied methods valid? Are the results discussed in an appropriate and balanced way (clarity of concepts and discussion, consideration of related work, including appropriate references)? | Excellent | **Good** | Fair | Poor |
| **3) Presentation quality** Are the scientific data, results and conclusions presented in a clear, concise, and well-structured way (number and quality of figures/tables, appropriate use of technical and English language, simplicity of the language)? | Excellent | **Good** | Fair | Poor |

For final publication, the manuscript should be

accepted as is.

**accepted subject to technical corrections.**

accepted subject to **minor revisions**.

reconsidered after **major revisions**:

rejected.

| Were a revised manuscript to be sent for another round of reviews: | |
| --- | --- |
| **I would be willing to review the revised manuscript.** | |
| I would not be willing to review the revised manuscript. | |
| | |

| **Suggestions for revision or reasons for rejection (will be published if the paper is accepted for final publication)** |
| --- |
| The authors answered all my comments and, in particular, added to the text the information regarding the computation of CHaines that was missing in the previous version of the manuscript and confirmed that data normalization was used for the MLP inputs. This information clears some doubt I had regarding these results in the previous version of the manuscript. Regarding my comment about the probabilistic approach, I agree with the authors that a refined dataset will allow for a better analysis, and I understand that such work can be a subject for future work. Despite the noisy dataset, the current results suggest that the FWI + EFBI provides a significant advantage over the FWI + CHaines for the discrimination between the two classes of fire size. This is a relevant result to the fire community and I expect it will be a good base for future research.

A few corrections:
L179 – "also CHaines index is also" I suggest removing the first "also".
L255 – "have" and "three cases"
L332 – EBI?
L339, L353 – These accuracy values are not consistent with the ones in the results section. |

**Authors: Changes applied.**

Anonymous Referee #2

| **Anonymous during peer-review:** | **Yes** | No |
| --- | --- | --- |
| **Anonymous in acknowledgements of published article:** | **Yes** | No |

| **Recommendation to the editor** | | | | |
| --- | --- | --- | --- | --- |
| **1) Scientific significance** **Does the manuscript represent a substantial contribution to the understanding of natural hazards and their consequences (new concepts, ideas, methods, or data)?** | Excellent | **Good** | Fair | Poor |
| **2) Scientific quality** **Are the scientific and/or technical approaches and the applied methods valid? Are the results discussed in an appropriate and balanced way (clarity of concepts and discussion, consideration of related work, including appropriate references)?** | Excellent | **Good** | Fair | Poor |
| **3) Presentation quality** **Are the scientific data, results and conclusions presented in a clear, concise,** | Excellent | **Good** | Fair | Poor |

| | |
|---|---|
| **and well-structured way (number and quality of figures/tables, appropriate use of technical and English language, simplicity of the language)?** | |

| |
|---|
| For final publication, the manuscript should be |
| **accepted as is**. |
| accepted subject to **technical corrections**. |
| **accepted subject to minor revisions.** |
| reconsidered after **major revisions**: |
| **rejected**. |
| |
| **Were a revised manuscript to be sent for another round of reviews:** |
| **I would be willing to review the revised manuscript.** |
| I would not be willing to review the revised manuscript. |
| |
| **Suggestions for revision or reasons for rejection** **(will be published if the paper is accepted for final publication)** |
| The authors have addressed all my concerns. The authors may wish to fine tune some of the English appearing in the revised text. |

**Authors: The article has been proofread-ed and corrections have been applied to improve the English.**